# Proteogenomic discovery of neoantigens facilitates personalized multi-antigen targeted T cell immunotherapy for brain tumors

Samuel Rivero-Hinojosa [1], Melanie Grant[1,2], Aswini Panigrahi[1], Huizhen Zhang[1], Veronika Caisova[1], Catherine M. Bollard [1,3] & Brian R. Rood[1,3 ✉]

Neoantigen discovery in pediatric brain tumors is hampered by their low mutational burden and scant tissue availability. Here we develop a proteogenomic approach combining tumor DNA/RNA sequencing and mass spectrometry proteomics to identify tumor-restricted (neoantigen) peptides arising from multiple genomic aberrations to generate a highly target-specific, autologous, personalized T cell immunotherapy. Our data indicate that aberrant splice junctions are the primary source of neoantigens in medulloblastoma, a common pediatric brain tumor. Proteogenomically identified tumor-specific peptides are immunogenic and generate MHC II-based T cell responses. Moreover, polyclonal and polyfunctional T cells specific for tumor-specific peptides effectively eliminate tumor cells in vitro. Targeting tumor-specific antigens obviates the issue of central immune tolerance while potentially providing a safety margin favoring combination with other immune-activating therapies. These findings demonstrate the proteogenomic discovery of immunogenic tumor-specific peptides and lay the groundwork for personalized targeted T cell therapies for children with brain tumors.

---

[1] Center for Cancer and Immunology Research, Children's National Research Institute, Washington, DC, USA. [2] Emory University School of Medicine, Department of Pediatrics, Atlanta, GA, USA. [3] George Washington University Cancer Center, Washington, DC, USA. ✉email: brood@childrensnational.org

Medulloblastoma (MB) is the most common malignant brain tumor of childhood and carries overall survival rates between 25 and 75% depending on the molecular subgroup, metastatic status, and age at diagnosis[1]. Effective therapy requires intensive chemo- and radiation therapies, leaving survivors with significant long-term burdens including life-altering cognitive deficits. If the tumor recurs after chemo-radiotherapy, there are no standard effective therapies and virtually no long-term survivors. Therefore, there is an urgent need to develop new therapeutics that can augment standard therapies to effectively prevent tumor recurrence without increasing toxicity. A promising strategy is the use of T cells targeting tumor-specific antigens (TSA), which can: (1) actively home to sites of disease, including across the BBB; (2) possess exquisitely sensitive peptide antigen recognition that may differ from their irrelevant counterparts by a single amino acid[2,3]; and (3) mediate continued, life-long protection through the generation of immune memory[4].

Strategies to expand T cell populations with specificity for multiple antigens expressed by a range of malignancies have been developed[5,6]. Targeting multiple antigens reduces the possibility of tumor resistance through antigen escape since selection pressure is not applied to a single target[7,8]. In addition, targeting multiple antigens more effectively addresses intra-tumoral antigen heterogeneity. Ex vivo expanded T cells targeting tumor-associated antigens (TAA) derived from differentiation antigens or cancer testis antigens have been evaluated in different cancer types[9–13]; however, these approaches may be limited by central tolerance toward antigens that are not wholly foreign as well as by the potential for on-target, off-tumor auto-immune toxicity[14]. When applied to solid tumors, additional challenges arise from the immunosuppressive tumor microenvironment. To enhance efficacy, it will be necessary to combine T cell therapies with immune adjuvants to boost immune activation and subvert the immunosuppressive tumor microenvironment. We propose that developing T cells targeting tumor-specific antigens (TSA), as opposed to tumor-associated antigens (TAA), will potentially increase the potency of tumor antigen-specific T cell products while decreasing the potential for toxicity, especially when administered in combination with immune adjuvants.

To identify sufficient TSA for multi-antigen targeting, it is necessary to expand their sources beyond somatic mutations alone. This is especially true for pediatric cancers which have many fewer mutations than their adult counterparts[15]. There are two main strategies to identify TSA. The first relies on the identification of non-canonical transcriptomic or exome sequencing reads followed by HLA binding prediction and large-scale immunogenicity assays[16,17]. However, this approach results in many false discoveries, relies on error-prone HLA binding prediction algorithms, and ignores the fact that many transcripts are not translated into proteins[18]. The second strategy is the direct immunoprecipitation of MHC-peptide complexes from tumor cells and identification of bound peptide sequences by liquid chromatography-mass spectrometry (LC-MS/MS) with subsequent matching to exome or transcriptomic reads. This strategy, called ligandomics, is limited by the efficiency of immuno-precipitation, a large amount of tissue required (infeasible for pediatric brain tumors), the need for robust tumor cell MHC expression, the identification of relatively few peptides, low throughput, and reliance upon digestion free spectra searches resulting in a higher degree of false positives[18,19].

To counter these limitations, we present a personalized, low-input (10−15 mg of tumor tissue) proteogenomic approach to identify TSAs resulting from an individual tumor's genomic aberrations and their use to manufacture T cells specific for multiple TSAs. We effectively identify TSAs arising from four types of genomic events: small insertions/deletions, single nucleotide variations (SNVs), fusions, and aberrant splice junctions. To verify that our findings are not simply unannotated normal proteins, we also developed a multi-step strategy to ensure that the tumor-specific peptides are not present in normal tissues. This approach succeeds in identifying a mean of tens of neoantigens per tumor, making multi-antigen targeting possible. As a proof-of-principle, we demonstrate that T cells selected and expanded in response to these peptides contain both CD4 and CD8 populations and are immunogenic, as demonstrated by cytokine profiling and robust cytotoxicity in vitro. We posit that this tool can be used to identify personalized TSA peptides for the creation of a T cell therapy using autologous TSA peptide-loaded dendritic cells to select and expand autologous T cells.

## Results

**Low-input proteogenomic workflow identifies multiple neoantigen peptides from individual medulloblastoma tumors.** Pediatric brain tumors present a difficult challenge for immunotherapy development given their low mutational burden, location behind the BBB in an immunosuppressive tumor microenvironment, intra-tumoral heterogeneity, and the frequently small amount of tumor tissue available for multiple diagnostic demands. We developed a low-input personalized proteogenomic approach for the identification and curation of tumor-specific neoantigens, which can be used to generate T cells, for autologous use, that are specific for multiple neoantigens in pediatric brain tumors (Fig. 1).

In order to identify tumor-specific genomic events, we obtained 46 freshly frozen tumor tissues and high coverage whole genome sequencing (WGS) and RNA-seq data from the Children's Brain Tumor Network (CBTN) (Supplementary Data 1). Four different types of tumor-specific genomic events were identified from genomic data: gene fusions, aberrant splice junctions, small insertions/deletions, and single nucleotide variants (SNVs). Gene fusions were called using three different algorithms: STAR-Fusion[20], ericscript[21] and Defuse,[22] with different numbers of intra- and inter-chromosomal fusions detected per tumor, a mean of 1,044 inter- and intra-chromosomal gene fusions per tumor were detected (Supplementary Fig. 1a). Coding SNVs were called on WGS using the GATK pipeline[23], detecting a mean of 0.8 coding mutations per Mb (Supplementary Fig. 1b). As expected, the number of SNVs indicates a low mutational burden as previously described for medulloblastoma[6]. Junctions were included if it was supported by more than 5 reads and was not present in the human Ensembl version 84 transcript annotation, detecting a mean of 12 junctions per Mb of the human genome (Supplementary Fig. 2). We detected a mean of 39,000 tumor-specific genomic events with the majority (37,000) belonging to the aberrant splice junction category (Fig. 2a). The tumor-specific genomic events for each tumor were translated into 1, 3, or 6 frames depending on the event and included in individualized databases (one for each tumor) together with the normal human proteome from UniProt (UP000005640, see "Methods" section for details).

MHC class I molecules typically present 8−10 amino acid peptides while MHC class II presents 11−30 amino acid peptides[24]. Our translational workflow employs autologous dendritic cells (DCs) to process tumor-specific peptides to the proper length and sequence for efficient MHC binding in the appropriate autologous HLA context. To test whether the lengths of the identified peptides were suitable to generate both MHC class I and II T cell responses after DC processing, we compared Lys-C and trypsin enzymatic digestion of the 7316-3778 tumor lysate. We also induced missed cleavages by reducing the

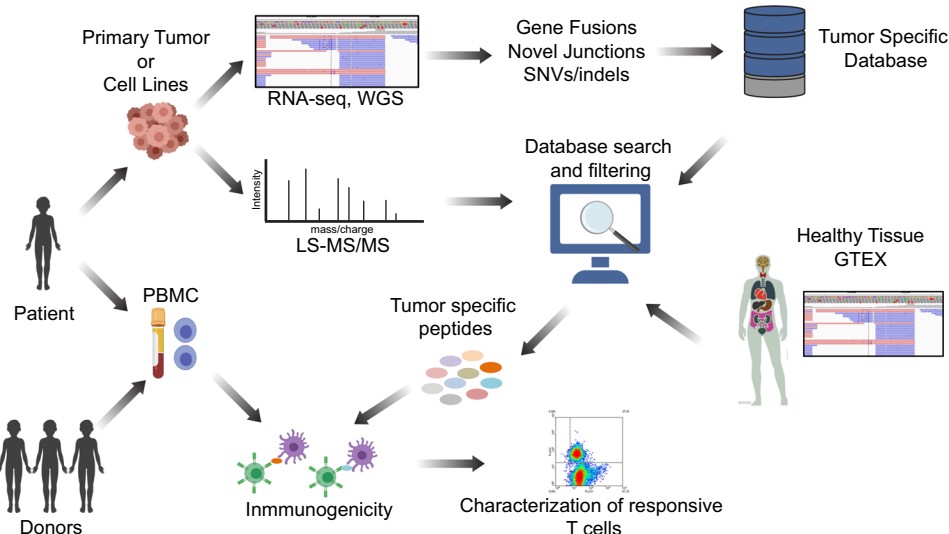

**Fig. 1 Proteogenomic approach to identify tumor-specific antigens in pediatric brain tumors.** Schematic representation of the entire workflow is shown. Tumor tissue samples were obtained from patients, and WGS and RNA-seq were performed to identify tumor-specific genomic aberrations (SNV/indels, junctions, and fusions). Protein lysates were subjected to LC-MS/MS shotgun proteomics and spectra were searched against tumor-specific databases originating from tumor WGS and RNA-seq. MS-identified peptides were filtered using genomic and proteomic data from normal tissues to eliminate potential non-annotated normal proteins. Finally, to evaluate tumor-specific peptides for immunogenicity, autologous and allogeneic T cells were selected and expanded against the peptides and characterized for phenotype and function.

digestion time to obtain longer average peptides, with an intent to yield more potential tumor-specific peptides suitable for MHC class I and II presentation after DCs processing. For this tumor, the Lys-C enzyme, which cleaves proteins only at lysine, resulted in peptides with a median of 17 amino acids in contrast to a median of 15 for trypsin digestion, which cleaves proteins at lysine and arginine (Supplementary Fig. 3a). These results indicate that Lys-C yielded longer peptides suitable to elicit both MHC Class I and II T cell responses after DCs processing.

We next performed high-resolution LC-MS/MS on the same panel of freshly frozen tumors for which we previously created individualized protein databases (Fig. 1 and Supplementary Data 1) using total protein lysates partially digested with LysC. In addition to the tumors, we also performed LC-MS/MS on five healthy childhood cerebellums for comparison. LS-MS/MS spectra from MB tumors and healthy cerebellums were searched together against the individualized tumor databases described above. Full and partial LysC digest (i.e., cleavage at a minimum of one C-terminal Lys) searches were performed. A total of 241,224 unique peptides were identified across the sample set with an average of 33,576 peptides per tumor with a 1% FDR (Supplementary Fig. 3b and Supplementary Data 2). In order to reduce the potential for false discovery in our proteogenomic findings, we developed a filtration strategy to remove peptides from annotated and unannotated normal proteins while preserving tumor-specific peptides from undiscovered protein-coding loci (Supplementary Fig. 4). We first removed all peptides matched to the human UniProt proteome and all unannotated peptides found in healthy childhood cerebellum. Additionally, peptides with a length < 8-mer and an Xcorr < 1 were culled. Xcorr (cross-correlation) is a measure of the goodness of fit of experimental peptide fragments to theoretical spectra created from the predicted b and y ions. Xcorr > 1 was selected based on our experience identifying confident matches between experimental and theoretical spectra. Next, the remaining identified peptides were processed through BLASTP[25] to remove exact matches to known proteins from the Human NCBI(GRCh38), RefSeq, UniProt Isoforms proteome (UP000005640), neXtProt, and Ensembl (version 84) protein annotations. Overall, 481

tumor-specific peptides were identified, and 17 (3.53%) peptides were removed as they were also present in normal cerebellar tissues (Fig. 2b, c). Finally, we removed peptides that originate from tumor-specific genomic events also present in normal healthy tissues from The Genotype-Tissue Expression (GTEx) project (Supplementary Data 1); 102 (21%) peptides were removed as the same genomic event that originates that peptide was detected in the GTEX RNA-seq collection. After these filtration steps, we identified a total of 362 unique tumor-specific peptides across all 46 MBs (Fig. 2b, c and Supplementary Data 3). A total of 230 peptides originated from splice junctions, 85 from SNVs, and 49 from gene fusions across all MB tumors analyzed. Thus, junctions are the main source of neoantigens identified in medulloblastoma tumors (Fig. 2b, c). On a per tumor basis, a mean of 9 tumor-specific peptides was identified (range 1−43) with an average of 5.9 originating from nonstandard junctions, 1.1 from gene fusions, and 1.8 from SNVs (Fig. 2b, c). The vast majority of these tumor-specific peptides were non-overlapping between the tumors. These results indicate that this approach identifies a significant number of high confidence tumor-specific peptides from minimal input tissue and that aberrant splice junctions are the main source of neoantigens in MB tumors[26].

**Tumor-specific peptides were validated by retention time and spectral match.** We employed three methods to evaluate the validity of the peptides discovered by our approach. First, we correlated the mean retention time (RT) of the identified peptides to their predicted hydrophobicity index (HI). The HI is a relative value that corresponds to the organic solvent concentration at which the peptide elutes from the HPLC system and is proportional to the retention time. The predicted HI for the UniProt matched peptides, calculated with the SSRCalc tool[27], shows a significant positive correlation with the experimental RT as shown in Supplementary Fig. 5 with a distribution along the regression line. Similarly, the predicted HI of the tumor-specific peptides has significant positive correlations with the retention times supporting their accurate amino acid sequence

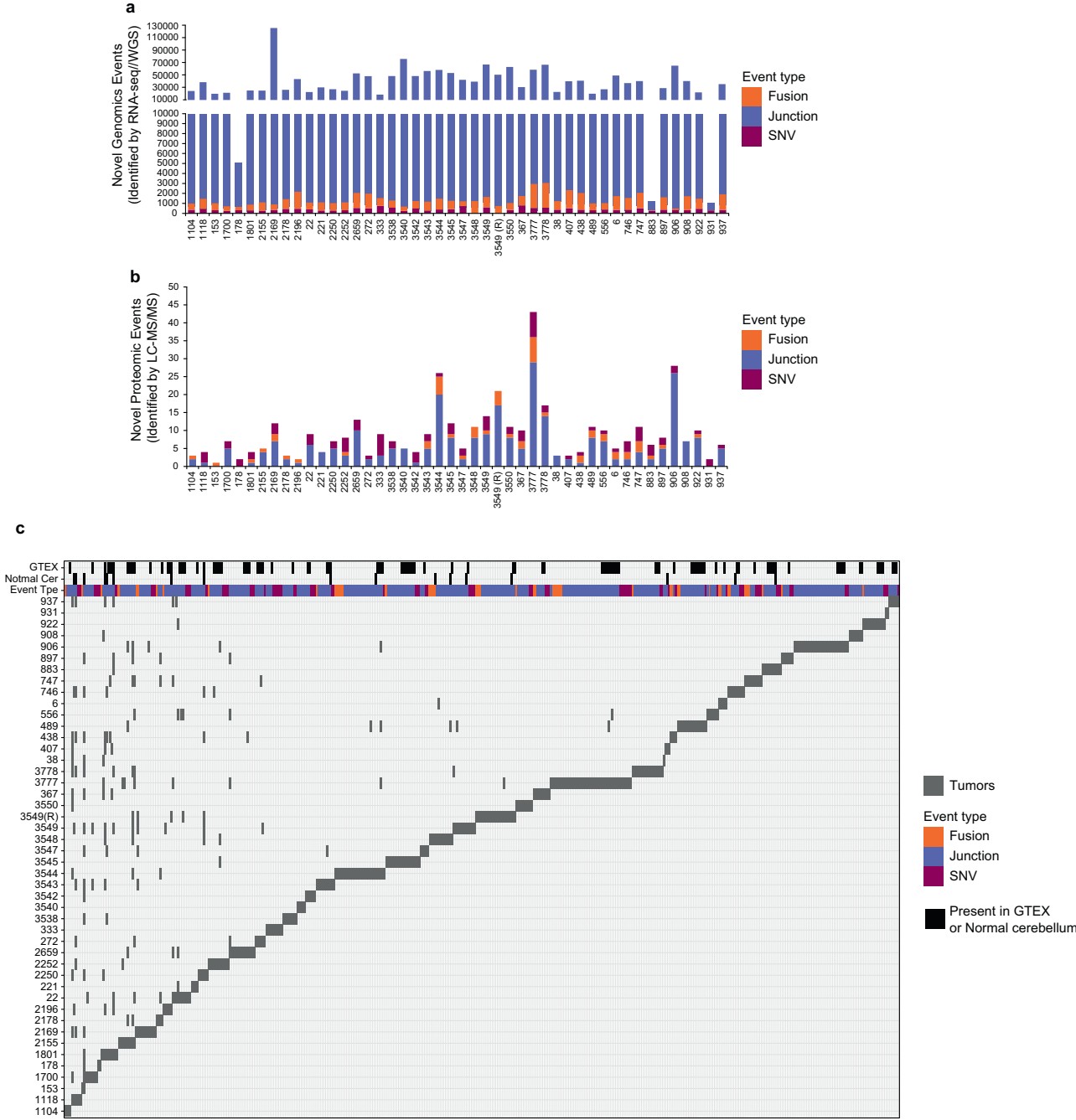

**Fig. 2 Tumor-specific genomic and proteomic events in 46 medulloblastoma tumors.** The number of **a** genomic (detected by RNA-seq/WGS) and **b** proteomic (detected by LC-MS/MS) tumor-specific events are shown for 46 medulloblastoma tumors. The type of tumor-specific genomic events is indicated: SNV/indels (purple), junctions (blue), and gene fusions (orange). The number of identified tumor-specific peptides ranged from 1 to 43 peptides per tumor with a mean of 9 peptides per tumor. **c** Tile plot depicting the number of tumor-specific peptides identified in medulloblastoma tumors. Black tiles indicate unannotated peptides identified in healthy cerebellum or normal tissues from GTEX. SNVs, fusions, and junctions are shown in purple, orange, and blue respectively. Each gray tile represents a peptide, although some peptides are found in multiple tumors, the vast majority are tumor-specific. Source data are provided as a Source Data file.

(Supplementary Fig. 5a). Second, we used AutoRT[28], a deep learning retention time prediction algorithm. A learning model was created using all peptides identified in our cohort (MB model) and applied to a subset of random identified uniport peptides and to all specific peptides (Supplementary Fig. 5b). We identified significant positive correlations (p-values shown in the Supplementary Fig. 5b) between predicted retention times and experimental retention times using the MB model for both

UniProt and tumor-specific peptides, (Supplementary Fig. 5b, left panel). We next used a completely independent learning model created using the PXD006109 dataset[29], to evaluate retention times of our peptides. Using this independent dataset, we found significant positive correlations (p-values shown in the Supplementary Fig. 5b) between predicted retention times and experimental retention for both UniProt and tumor-specific peptides (Supplementary Fig. 5b, right panel). Third, synthetic versions of

7 tumor-specific peptides from patients 7316-3778 were analyzed and, as shown in Supplementary Fig. 6, their MS/MS spectra and retention times were identical to those detected endogenously (Supplementary Fig. 6). These results indicate that our proof-of-principle proteomic approach properly identifies MB tumor-specific peptides.

**Tumor-specific peptide frequency**. Although most of these tumor-specific peptides were non-overlapping between the tumors, 18 of 362 (4.97%) were identified in more than one tumor; 12 of them (3.3%) were found in six tumors and six of them (1.6%) were identified in five or more tumors (Supplementary Fig. 7). Only one peptide, NSSVSGIFTFQK, was identified in more than 20% (12 of 46) of the samples. All of these shared peptides result from aberrant splice junctions. The peptide NSSVSGIFTFQK can arise from a number of different aberrant splice junction events in the *DDX31* gene. We detected alternative splicing between the exon 14 and exons 17, 18, and 19. In addition, we detected an aberrant splice junction between intron 13 and exon 14. This junction changes the frame of exon 14, originating the peptide NSSVSGIFTFQK. Interestedly, The splicing between exon 14 and exon 17, 18, and 19 returns to the canonical frame for the DDX31 protein. The annotated splicing between the Exon 14 and 15 will introduce stop codons as consequences of this change in the frame of exon 14.

DDX31 is a DEAD-box RNA helicase conserved across eukaryotes, but it has not been extensively studied. *DDX31* was found to be mutated in several group 4 medulloblastomas in a previous sequencing study[30]. Complex rearrangement and focal deletions of the *DDX31* gene have also been observed in several Group 4 medulloblastomas; these deletions occur concurrently with amplification of the *OTX2* locus, a known medulloblastoma oncogene[31,32]. This finding suggests that *DDX31* mutation (either by deletion or truncation) may cooperate with the oncogenic role of OTX2. Linking our findings to previously published work, such deletion or rearrangement could originate the NSSVSGIFTFQK peptide by modifying the splicing pattern of DDX31. In addition, peptides arising from aberrant splice junctions in the genes *CARF, EEA1, LMNB1, LIZIC,* and *VANGL2* have been discovered with lower frequency in 5 out of 46 tumors (Supplementary Fig. 7).

**TSA-specific T cells respond to medulloblastoma tumor-specific neoantigens**. As a further proof-of-principle to determine the clinical feasibility of using this neoantigen identification approach to create an autologous T cell immunotherapy product from heavily pre-treated patients, we identified a subject (Patient ID: 7316-3778) from whom tumor tissue and blood were available. Employing our proteogenomic pipeline, we identified 25 tumor-specific unique peptides. Among these 25 peptides, three peptides were also identified in the healthy cerebellum proteome. In addition, we discovered that the genomic events giving rise to six tumor-specific peptide sequences were present in normal tissues from GTEX. After removal of those normal unannotated peptides, we were left with 17 unique peptides derived from this subject's tumor sample. One peptide resulted from a fusion, two from SNVs and 14 from aberrant splice junctions (Fig. 3a).

To create an autologous Tumor-Specific Antigen T cell product (TSA-T) specific for this patient's newly identified tumor-specific peptides, we stimulated their peripheral blood mononuclear cells (PBMC) with antigen-presenting cells (dendritic cells; DC) pulsed with 15 out of 17 peptides derived from patient 7316-3778's tumor (two peptides could not be synthesized; Supplementary Data 4). Peptide-pulsed DC stimulation was repeated on days 7 (2nd stimulation) and day 14 (3rd stimulation). Seven days after the 3rd stimulation, the polyclonality and polyfunctionality of the resultant T cells were evaluated using anti-interferon gamma (IFN-γ) ELISpot assay and intracellular flow cytometric staining for IFN-γ and tumor necrosis factor-alpha (TNF-α). TSA-T cell products stimulated with DCs loaded with 13/15 peptides elicited a statistically significant IFN-γ response (Fig. 3b). This response was reproducible even when using cryopreserved, thawed, and rested TSA-T cells as opposed to fresh product.

The TSA-T cell product derived from patient 7316-3778 was polyclonal comprising 93% CD3+, 31% CD8+, 29% CD4+, 9.42% NKT, and 2.11% NK cells. The differentiation and memory status included primarily TEM (CD4+: 44%, CD8+: 21%) and TCM (CD4+: 16%, CD8+: 44%) with minimal TEFF and TSCM populations (Fig. 3c). TSA-T cells primed and expanded with autologous tumor-specific peptides comprised approximately equal proportions of CD8+ (31%) and CD4+ T cells (29%). TSA-T cells showed polyfunctionality producing IFN-γ, TNF-α, or both in response to peptide-loaded DCs. Polyfunctional responses were observed in both CD8+ (IFN-γ: 3.1%; TNF-α: 2.71%; IFN-γ+ TNF-α+: 1.55%) and CD4+ cells (IFN-γ: 3.25%; TNF-α: 5.70%; IFN-γ+/TNF-α+: 6.66%). As expected, CD8+ and CD4+ Tcells did not produce TNF-α and/or IFN-γ in the presence of peptides that were not presented by DCs (Fig. 3d and Supplementary Fig. 8). Thus, antigen-presenting cells pulsed with TSA peptides can prime and expand autologous T cells comprising a polyfunctional polyclonal population of CD4 and CD8 cells with robust TEM and TCM fractions (Fig. 3d and Supplementary Fig. 8).

Conventional assays such as anti-IFN-γ ELISpot and intracellular flow cytometry for detecting tumor-specific T cell responses in cancer patients can underestimate the breadth of antigen-specific T cell responses, and do not assess antigen-specific T-cell repertoires[33]. Therefore, we also performed T cell receptor Vβ sequencing to evaluate the expansion of autologous antigen-specific clones. Dominant clones were identified with the top TCR clone accounting for 18% of all unique rearrangements and the top 10 clones making up 43.77% of all TCRs (Fig. 3e and Supplementary Data 5). Thus, in combination with the ELISpot data, TCR sequencing further supported that the presence of multiple specifically expanded clones within this patient's TSA-T cell product.

**TSA-specific T cells can be generated from medulloblastoma cell line-derived neoantigens**. The preceding findings demonstrate a proteogenomic approach to identifying tumor-specific peptides and their ability to select and expand a mixed lineage, multi-functional, multi-antigen specific autologous TSA-T cell product. The last remaining measure of activity to be demonstrated is the ability of TSA-T cell products to recognize and lyse tumor cells. Because there was no autologous tumor cell line was generated from this subject's tumor tissue, it was necessary to replicate the previous steps using in vitro MB cell lines. Using four MB cell lines (D556, MB002, MB004, and D283), we generated high coverage RNA-seq data from poly-A RNA and three distinct types of tumor-specific genomic events were identified from genomic data: gene fusions, aberrant splice junctions, and small insertions/deletions (SNVs). We detected a mean of 67,000 tumor-specific events with the majority (59,000) of them belonging to the aberrant splice junction category (Fig. 4a). We detected a mean of 6,000 inter- and intra-chromosomal gene fusions per cell line, 0.7 coding mutations per Mb, and 20 aberrant splice junctions per Mb of the human genome (Supplementary Fig. 9a−c). These genomic events were translated into 1, 3, or 6 frames depending on the event and included in cell line-specific databases together with the normal human proteome from UniProt (see "Methods" section for details).

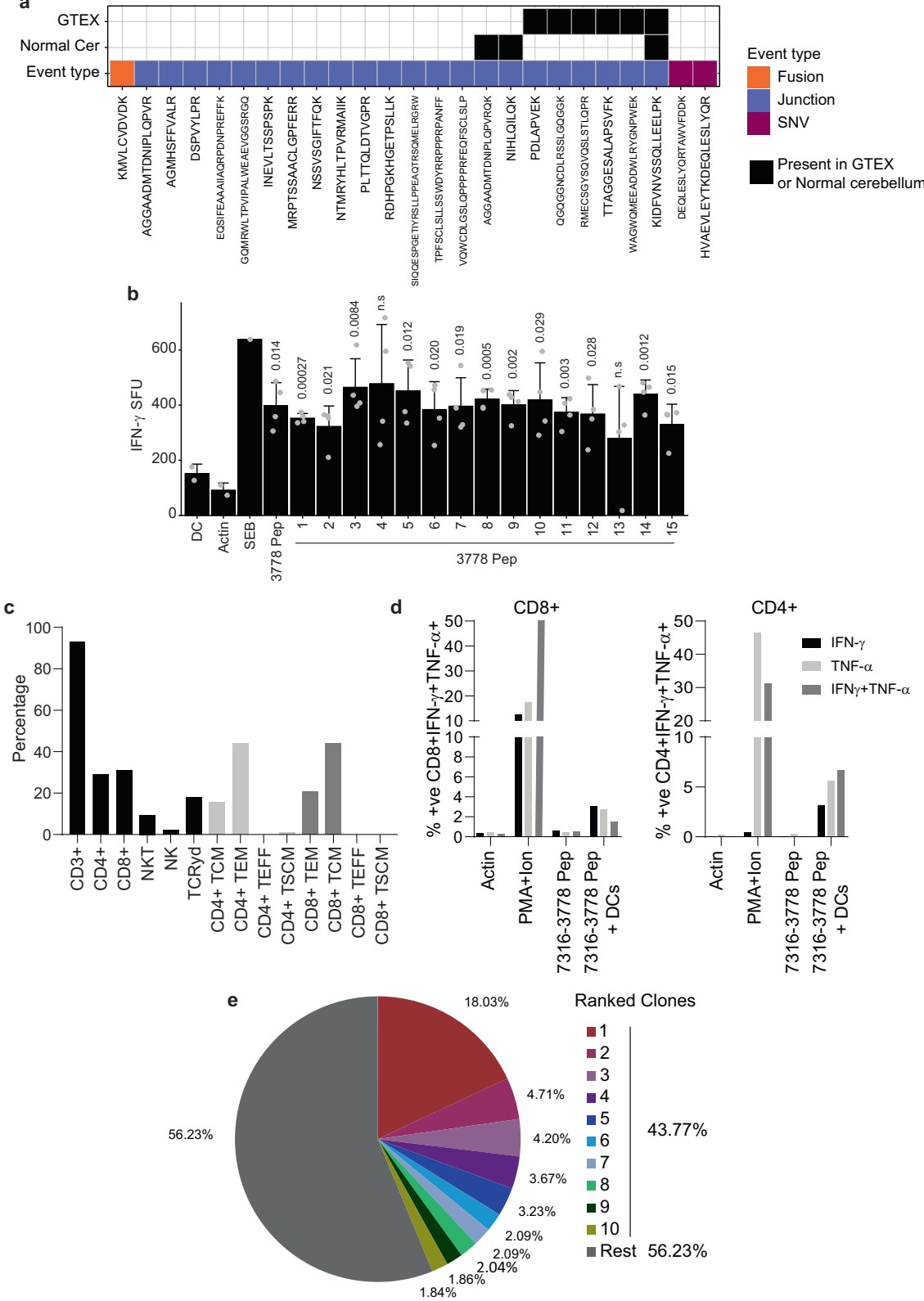

Similar to the approach used for identifying TSAs from primary MB tumor tissues, we performed global high-resolution peptide LC-MS/MS on the MB cell lines using total protein lysates after protease digestion. Applying the same proteogenomic cohort described for MB tumors a total of 100,771 unique peptides were identified across datasets with an average of 53,649 peptides per cell line with a 1% FDR (Supplementary Fig. 9d and Supplementary

Data 6). The removal of normal peptides (from UniProt and our BLASTP approach) and unannotated peptides found in the healthy cerebellum proteome reduced the number to 53, 75, 74, and 132 tumor-specific peptides remained for the D556, MB002, MB004, and D283 cell lines, respectively (Fig. 4b, c). Finally, we removed peptides that originate from genomic events also present in normal healthy tissues from The Genotype-Tissue Expression

**Fig. 3 Autologous tumor-specific peptides induce a specific IFN-γ response. a** Tile plot depicting the number of tumor-specific peptides identified in patient 7316-3778's tumor. Black tiles indicate the unannotated peptides identified in healthy cerebellar or normal tissues from GTEX. SNVs, fusions, and junctions are displayed in purple, orange, and blue respectively. **b** Dendritic cells derived from patients 7316-3778 were loaded with peptides identified in the subject's tumor by our proteogenomic pipeline. DCs were co-cultured with non-adherent cells as described. Following three stimulations, peptide-specific responses were assessed by anti-IFN-γ ELISpot. In the presence of peptide-loaded DCs, a significant anti-IFN-γ response was observed against 13/15 peptides. One-sided t-test was used to calculate the p-values, n = 4. p-values are indicated in the figure, n.s. non-significant. **c** Summary data of patient TSAT phenotype, memory, and differentiation status. Gating strategy TSAT populations and phenotypes (Supplementary Fig. 15). **d** To assess CD4- and CD8-specific cytokine function, TSAT were incubated in the presence of pooled peptides or peptide-loaded DCs. Summary of intracellular staining data showing specific CD4+ and CD8+ responses to 7316-3778 peptides in the presence of peptide-loaded DCs. Gating strategy for intracellular staining (Supplementary Fig. 15). **e** Results of TCR Vβ CDR3 sequencing on the 7316-3778 TSA T product. Pie chart of the top 10 clonotypes in the TSA T. Clonotypes are listed in Supplementary Data 5. SFU spot-forming units; 1 SFU = 1 T cell secreting IFN-γ. Actin: specific peptide control; PMA/Ionomycin positive control. Data are presented as mean values +/− SD. Source data are provided as a Source Data file.

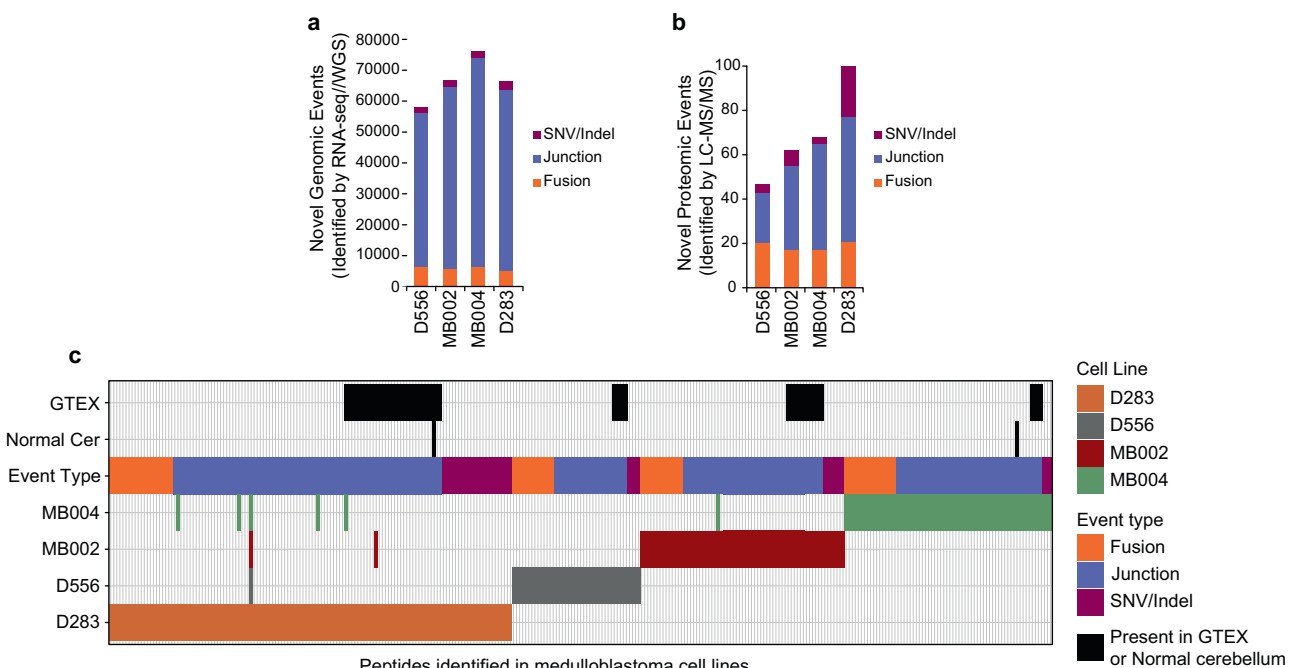

**Fig. 4 Tumor-specific genomic and proteomic events in medulloblastoma cells lines.** The number of **a** genomic (detected by RNA-seq/WGS) and **b** proteomic (detected by LC-MS/MS) tumor-specific events are shown for D556, MB002, MB004, and D283 medulloblastoma cell lines. The type of tumor-specific genomic events is indicated: SNV/indels (purple), junctions (blue), and gene fusions (orange). **c** Tile plot depicting the number of tumor-specific peptides identified in medulloblastoma cell lines. Black tiles indicate unannotated peptides identified in healthy cerebellum or normal tissues from GTEX. SNVs, fusions, and junctions are shown in green, red, and blue respectively. Source data are provided as a Source Data file.

(GTEx) project (Supplementary Data 1 and Fig. 4c). We identified a total of 269 unique tumor-specific peptides across the four MB cell lines—47, 62, 68, and 100 peptides in the D556, MB002, MB004, and D283 lines, respectively (Supplementary Data 7). The vast majority of these were non-overlapping between the cell lines. Moreover, no peptides were found to be in common between cell lines and primary tumor tissues. In regard to the types of genomic events giving rise to tumor-specific peptides, 157 originated from aberrant splice junctions, 37 from SNVs, and 75 from gene fusions (Fig. 4b, c). These results indicate that aberrant splice junctions are also the main source of neoantigens in MB cell lines.

We then investigated whether neoantigens discovered in the MB002 and D556 cell lines using our proteogenomic strategy would be recognized as immunogenic by HLA-matched donor-derived T cells (Supplementary Data 8). We stimulated these donor-derived T cells with DCs derived from the same donor and pulsed with the tumor-specific peptides identified from the MB002 or D556 MB cell lines (Supplementary Data 9). We validated these MB cell line-specific peptides against synthetic versions and showed that their MS/MS spectra and elution times

matched those found in the cell lines (Supplementary Figs. 10 and 11). Seven days after the 3rd stimulation, the TSA-T cells were re-stimulated with DC pulsed with MB cell line-specific peptides and assessed by anti- IFN-γ ELISpot and intracellular flow cytometric staining for IFN-γ and TNF-α) TSA-T cells stimulated with DCs pulsed with peptides discovered from the MB002 cell line were evaluated in three healthy donors. In Donor 1, a significant IFN-γ T cell response to the pool of all the MB002 peptides was observed (p-value = 2.01e−4 compared to actin and p-value = 1.97e−4 compared to unrelated D556 peptides), while no response was observed to DMSO, Actin, or unrelated (D556 pep) peptides (Fig. 5a). To determine whether this response was MHC-restricted, we incubated MB002 TSA-T cells with pooled MB002 peptide-pulsed DCs in the presence of anti-MHC Class I or class II blocking antibodies. No reduction was observed in the presence of anti-MHC Class I antibody. A significant reduction (80%, p-value = 6.29e−4) occurred in the presence of anti-MHC Class II antibody (Fig. 5a), which correlated with the autologous data that also demonstrated a predominant CD4+ restricted TSA-specific T cell response. To assess the individual

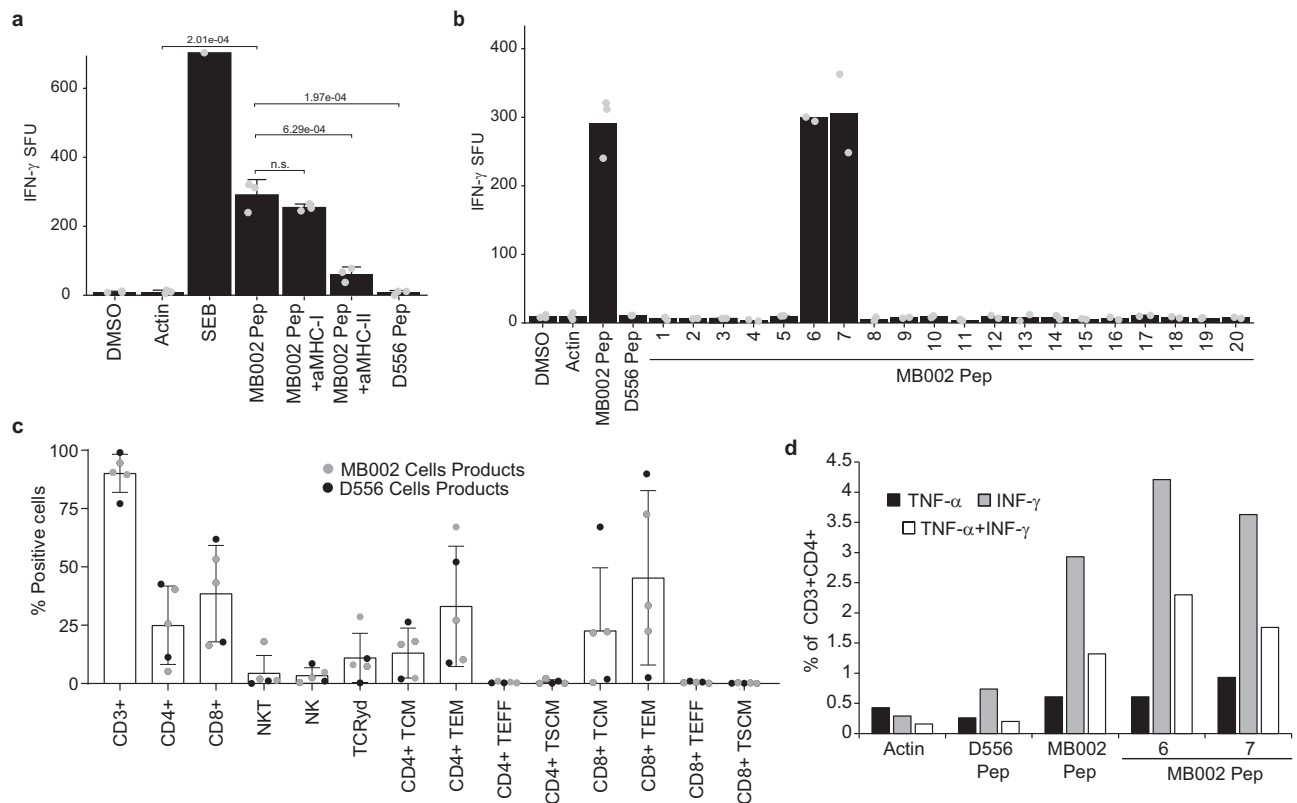

**Fig. 5 MB002 cell line-specific peptides induce peptide-specific, poly-functional Class II-mediated T cell IFN-γ responses in vitro.** Dendritic cells were loaded with MB002-specific peptides and co-cultured with non-adherent PBMCs. After three stimulations, peptide-specific T cell responses were analyzed by anti-IFN-γ ELISpot. **a** Summary data of IFN-γ response to pooled MB002 peptides in healthy donor 1. One-sided t-test was used to calculate the p-values, n = 3. p-values are indicated in the figure. **b** Summary data showing IFN-γ response to individual MB002 peptides in healthy Donor 2 (n = 2). **c** Summary data of MB002 and D556 TSATs populations and phenotypes (Error bars: mean + SD of five independent experiments). **d** Summary data showing CD4+ cytokine response to the pooled MB002 peptides and to the individual peptides 6 and 7 in the healthy donor 2. SFU: spot-forming units; 1 SFU = 1 T cell secreting IFN-γ; TSA T: Tumor-specific antigen T cell; a-MHC-I/II: anti-MHC Class I/II blocking antibodies; MB002 pep: pooled MB002 peptides; D556 pep: pooled D556 peptides; DMSO: dimethyl sulfoxide (peptide solvent; unstimulated control); actin: peptide specificity control; SEB: staphylococcus enterotoxin B (positive control). Gating strategy for intracellular staining (Supplementary Fig. 15). Gating strategy TSA T populations and phenotypes (Supplementary Fig. 16). Data are presented as mean values +/− SD. Source data are provided as a Source Data file.

contribution of each peptide to the T cell response, we plated MB002 TSA-T cells with DCs loaded with each individual TSA peptide. In Donor 2, a response to peptides 6 and 7 was observed, similar in size to the pooled peptides (Fig. 5b and Supplementary Fig. 12b). Lower frequency, but specific responses to peptides 6 and 7 were also observed in healthy Donor 3-derived TSA-T (Supplementary Fig. 12a).

We further mapped the MHC class II-restricted responses identified in TSA-T cells that had been generated from healthy Donor 1 using DC pulsed with MB002-derived peptides. TSA-T were stimulated with DCs pulsed with mini pools (3−4 peptides/pool) in the presence of anti-MHC class I and class II blocking antibodies (Supplementary Fig. 12c, d). In the presence of MHC-II-blocking antibodies, the frequency of SFU was reduced by approximately 50% in peptide pools 7−10. A similar reduction of 50% was also observed in pools 4−6. In contrast, in the presence of MHC I-blocking antibodies the frequency of SFU in peptide pools 4−6 and 7−10 was not reduced. These data further confirmed that these peptides were recognized in the context of MHC class II mimicking the autologous result. Only the pools containing peptides 6 (pool 4−6) and 7 (pool 7−10) showed robust responses, which were reduced by anti-MHC class II, confirming that this class II-restricted epitope spanned peptides 6 and 7 corresponding to sequence KASELDYITYLSIFDQLFDIPK (Fig. 5A and Supplementary Fig. 12c, d).

To evaluate reproducibility beyond a single tumor cell line, TSA peptides identified from the D556 MB cell line were tested in the same way as MB002-derived TSA peptides in two healthy donors (Donors 4 and 5). IFN-γ release, indicating a positive T cell response, was likewise observed for 2 and 1 (out of 21) D556 peptides in healthy Donor 4 (Supplementary Fig. 13a) and healthy Donor 5 (Supplementary Fig. 13b) derived TSA-T products respectively. Together these results demonstrate that tumor-specific peptides identified in MB002 and D556 MB cell lines were able to prime and stimulate TSA-specific T cells in a partially HLA-matched allogeneic setting.

The phenotype of expanded TSA-T products stimulated with MB002 or D556 tumor cell line-derived peptides was evaluated by 11-colour flow cytometry. All populations comprised primarily CD3+ T cells (median: 91%; range: 77−95%) with variable compositions of CD8+ T cells (37%; 17−55%), CD4+ T cells (23.5%; 5−40%), NKT cells (37%; 18−64%), TCRγδ cells (8%; 0.3−31%), and NK cells (1.85%; 0.46−8.5%) (Fig. 5c). The differentiation and memory status were likewise variable across donors with T effector memory (TEM) populations with a range of CD8+ (2.5−91%) and CD4+ (8.9−67%) T cells. The NKT cell proportions reflect our experience with clinical trial products manufactured using the same methods to generate tumor-associated antigens (TAAs)[34]. T central memory (TCM) cells, shown to be important for long-term persistence of adoptively

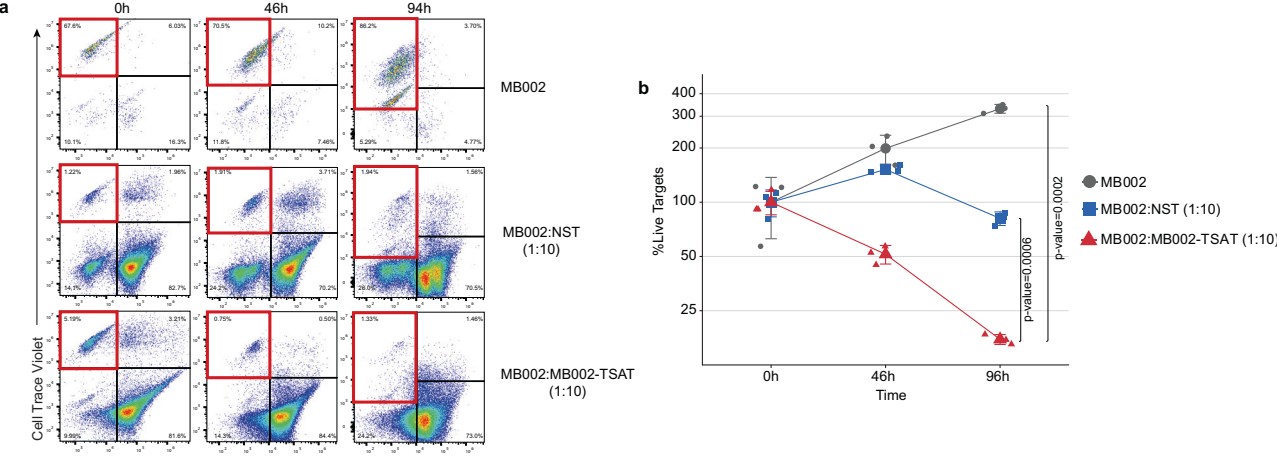

**Fig. 6 MB002 TSA T cells specifically lyse partially HLA-matched tumor cells.** To assess cytotoxic function, cryopreserved TSA T were thawed and rested overnight prior to plating with tumor targets. At the indicated time points, co-cultures were harvested and acquired as described in the "Methods". **a** Representative dot plots from one healthy donor showing proliferation of tumor targets in the absence of TSA T (top row), moderate reduction of tumor targets in the presence of non-specifically activated T cells (PHA blasts; middle row), and robust lysis of tumor targets in the presence of TSA T (bottom row). Lysis was determined based on the disappearance of targets from quadrant 1 (red border). **b** Summary data of (**a**). NST non-specific T cells (PHA blasts). One-way ANOVA p-values are shown. Values at each time point were normalized to 0 h (100%). One-sided ANOVA test was used to calculate the p-values, n = 3. p-values are indicated in the figure. Gating strategy for expanded TSA T tumor cell cytotoxicity assays shown in Supplementary Fig. 18. Data are presented as mean values +/− SD. Source data are provided as a Source Data file.

transferred T cells in vivo[35], also varied greatly between donors (CD8+ range: 0.43−67%; CD4+ range: 1.9−26%). Minimal numbers (<1%) of T effector cells (TEFF) or stem cell memory T cells (TSCM) were detected (Fig. 5c).

To evaluate the polyfunctionality (TNF-α and IFN-γ-production) of TSA-T cells derived from Donor 2, we stimulated TSA-T with MB002 peptides and intracellularly labeled them with antibodies against IFN-γ and TNF-α (Fig. 5d and Supplementary Fig. 14a). In agreement with the ELISpot (Fig. 5a), Donor 2's CD4+ T cells produced IFN-γ &/or TNF-α in response to pooled MB002 peptides, and peptides 6 and 7 (7.1 and 6.3% respectively; Fig. 5d and Supplementary Fig. 14a). Thus, the intracellular cytokine data corroborated the HLA-blocking ELISpot results, supporting the evidence that the epitope spanning peptides 6 and 7 (sequence KASELDYITYLSIFDQLFDIPK) is class II-restricted.

Similarly, following stimulation with D556 cell line-specific peptides, CD4+ T (but not CD8+) cells derived from healthy donor 4 secreted IFN-γ and TNF-α in response to peptides 6 (2.95%) and 20 (4%) (Supplementary Fig. 14b, c). In summary, multiple CD4-mediated responses were observed in healthy donor-derived TSA-T cell products primed and expanded with DCs pulsed with peptides from MB002 and D556 cell lines. Moreover, these data confirm that we have established a robust approach to generate polyclonal and polyfunctional TSA-T cells products recognizing tumor-specific peptides identified using our proteogenomic strategy.

**TSA-T cell products are cytolytic against tumor targets in vitro**. The cytolytic function of healthy donor-derived TSA-T was evaluated against the MB002 cell line (Fig. 6 and Supplementary Fig. 17). The TSAT product used in this assay was derived from Donor 1 which matched the MB002 cell line at 4 HLA alleles (Supplementary Data 8). Despite low expression of MHC Class I and Class II on MB002 cells (Supplementary Fig. 19), TSA-T induced significant lysis of MB002 cells, with only 15% of MB002 cells remaining after 94 h co-culture compared with non-specifically activated T cells (NST; PHA blasts) (80%, p-value = 0.0006) and untreated MB002 cells (287%, p-value = 0.0002) (Fig. 6a, b). To demonstrate the reproducibility of this approach, the assay was

repeated in a subsequent experiment using a TSA-T product derived from a different donor (Donor 2), which also matched at 4 HLA alleles, including HLA-C, DRB1, and DRB5 (Supplementary Data 8). After 96 h of co-culture with TSA-T, MB002 cells were reduced to 7% of the original population, compared to 57% in the wells cultured with non-specific T cells (NST) derived from the same donor (Supplementary Fig. 17a, b). Thus, in two independent experiments, evaluating TSA-T products derived from two different donors, the specific cytotoxic capacity of the peptide-primed TSA-T was observed, which outperformed that of a non-specific, potentially allogeneic response induced by NST.

## Discussion

Current chemoradiotherapy treatments for malignant pediatric brain tumors suffer from insufficient efficacy coupled with significant lifelong sequelae for survivors[36]. Both constitute failures for which innovative solutions need to be envisioned. T cell-based therapies represent a class of interventions with attributes suited to the treatment of brain tumors. T cells efficiently penetrate the blood-brain barrier, one of the main challenges for drug development[37–39]. They can home to areas of disease to eliminate small amounts of residual tumor. And unlike conventional therapies, T cells possess an exquisite ability to distinguish tumors from normal cells, thus eliminating a primary source of toxicity[3,40]. However, T cell therapies also have several challenges to be overcome before they can become a standard of care. Solid tumors create an immunosuppressive microenvironment characterized by exhaustion-inducing checkpoint molecules, anti-inflammatory cytokines, and inhibitors of T cell migration[8]. In addition, the tumor cells themselves can downregulate MHC expression to limit antigen presentation, as well as downregulate the expression of antigens in order to escape immunosurveillance. Combinatorial immunotherapy approaches will be required to overcome these obstacles.

In considering immunoadjuvant therapies designed to magnify T cell responses, it is important to think about why these systems evolved in the first place. An activated immune system is a dual-edged sword with a fine balance to be maintained: too little and the host is unprotected from infection or invasion, while too

much activity results in life-threatening inflammation. Further, inappropriate non-specific activity causes autoimmune disease. Immunoadjuvants are relatively blunt tools and thus their application does not represent an increase of the therapeutic index but rather an amplification of the cellular response, for both good and ill. Therefore, the burden of specificity lies with the T cell backbone to which these magnifiers will be applied.

Particular attributes of T cell therapies will be required to increase efficacy while decreasing off-target effects. Indeed, improved efficacy itself is a safety feature in that the more effective the T cell employed, the fewer non-specific immunoadjuvant interventions will be required. First, multi-antigen targeting will be critical to target all subclones in a heterogeneous tumor, while also making it more difficult for the tumor cell to down-regulate the targeted antigens[7]. Epitope spreading will be an important feature of a successful T cell response[41,42], but the likelihood of it taking place will depend on the magnitude of the initial immuno-activation and tumor cell lysis. Second, targeting antigens that are completely specific to the tumor and expressed nowhere else in the body both increases safety and avoids the need to overcome the central immune tolerance that protects most endogenous proteins[43]. Third, autologous T cell products can react to antigens presented by all resident HLA alleles, due to the complete HLA match between tumor and T cells from the same host, thus eliminating disparities of access inherent in off-the-shelf HLA-restricted products such as peptide vaccines. In addition, T cells expanded from an autologous source likely include cells that have been exposed to the tumor's antigens, although ineffectively, and thus the manufacturing process can benefit from a memory response to those antigens. In our data, the TSA peptides identified in primary tumor cells elicited more robust and ubiquitous immunogenicity by ELISpot than those identified in the cell lines. The explanation for this could include the complete HLA matching and presence of memory T cell responses in the autologous as opposed to the allogenic settings. Fourth, we show that the majority of tumor-specific neoantigens are private to an individual tumor, necessitating a personalized process for neoantigen identification and T cell production. This personalization could also potentially enhance the specificity of the T cell product.

The proteogenomic neoantigen identification workflow presented here represents a refinement over previously published approaches with an intent to mitigate some of the challenges inherent in T cell therapy for pediatric brain tumors. The most mature approaches combine elements of affinity capture of MHC complexes from tumor cells with subsequent LC-MS/MS of bound peptides and searching peptide spectra against custom peptide databases containing aberrant events (usually mutations) present in tumor genomic sequencing[44]. Affinity capture of MHC complexes from tumor cells requires a significant amount of tissue, far more than is available from a typical pediatric brain tumor resection. In contrast, our approach requires a very low input of tumor tissue, around 10 mg. Further, the limited efficiency of immunoprecipitation results in very few identified peptides, which hampers the ability to target the multiple antigens necessary to deal with tumor heterogeneity and antigen escape. Our approach yields an average of 9 peptides per clinical tumor sample and those peptides originating from non-coding sequences can be used to predict multiple other tumor-specific peptides. Our workflow identifies longer tumor-specific peptide substrates and relies on the native peptide processing machinery of autologous dendritic cells to present the optimal epitope for T cell selection and expansion; longer peptides preserve the potential for MHCII presentation and CD4+ T cell activation. We make use of custom databases for peptide searches but unlike most preceding efforts, our pipeline incorporates aberrant splice junctions, SNVs,

and fusion events rather than just the more typical coding mutations. This is necessary because pediatric brain tumors, and pediatric cancers in general, have among the lowest mutation rates of all cancers[15]. Aberrant splice junctions have been predicted to be a source of neoantigens, but this is the first demonstration of their dominance in primary tumor samples[26]. These aberrant splice junctions occur in six different types (see "Methods") with the most common source of tumor-specific peptides being junctions joining 2 non-exon regions (40%). Because some proteomic studies have reported the detection of peptides from non-coding RNAs and their potential use as neoantigens[45,46], we investigated to determine if junctions involving non-coding RNAs could explain a portion of our tumor-specific peptides. We found that approximately 20% of junctions involving non-exon regions originate from non-coding RNA transcripts found in the Noncodev6 database[47], linking our findings to previous work.

Our workflow does have limitations. Firstly, MS identification of tumor-specific peptides demonstrates their presence in the cell but does not verify their surface presentation. Second, we have attempted to augment the activation of CD4+ T cells but the proportion of CD4+ and CD8+ T cells activated by TSAs needs to be further investigated in a larger cohort, preferably in the context of a GMP supported the clinical trial. Third, TSA peptides arising from unstable translation may be quite short and lack a necessary lysine residue to allow for LysC digestion and subsequent identification in proteomic searches. In addition, if LysC digestion results in products less than 6-mer, these will be excluded by our MS filtration steps. This is not a limitation of ligandomics where digestion is not performed. However, digestion-free searches have a higher degree of false discoveries. Fourth, our workflow incorporates multiple filtration steps in order to screen out normal unannotated peptides, though it cannot be claimed that this process will detect all such peptides. We believe this to be a minor limitation in the context of antigen-specific T cells insofar as tumor-associated antigens (TAAs) have been targeted using this platform without appreciable toxicity[34]. TAAs are expressed in limited tissues or at low levels somatically and thus targeting them would be expected to be analogous to targeting an unannotated normal protein. In addition, central tolerance would be expected to cull any high-affinity T cell receptors for such normal proteins. Despite these limitations, the pipeline presented creates an alternative and/or complimentary approach to ligandomics.

Our data demonstrate specific CD4+-restricted responses on ELISPOT. The well-characterized anti-tumor role of CD8+ T cells has long been prioritized for immunotherapy; however, the advantages of transferred cytolytic CD4+ T cells to immune activation are receiving more attention[48–50]. Most commercial peptide mixes for neoantigen adoptive cellular therapy contain 9-mer peptides and are thus optimized for MHC I-presented CD8+ epitopes. The partial enzymatic digestion strategy in our mass spec workflow is designed to identify long peptides which, when processed by autologous dendritic cells into immunogenic epitopes, should increase the selection and expansion of CD4+cells. Several publications have shown that DCs can efficiently intake, process, and present long synthetic peptides[51–54]. Our results indicate the ability to engage CD4+ T cells, particularly for the TSA T cells targeting MB cell line-derived TSAs, which appear to elicit a preponderance of CD4+-restricted responses. Indeed, the TSA peptides that showed the strongest activation by ELISpot acted through MHC II.

The uniqueness of the majority of the tumor-specific tumor peptides we identified indicates that the genomic events that generate these peptides are unlikely be tumorigenic. We cannot disregard the possibility that some of these peptides may

contribute to tumor formation as very few high-frequency driver events have been identified in medulloblastoma tumors despite intensive genomic study[55,56]. Commonly held driver events in medulloblastoma occur at a relatively low frequency compared with other cancers, particularly adult cancers. The DDX31 finding discussed above may be one such event given the multiple ways in which it can be perturbed—mutation, deletion, and now aberrant splicing[30]. Furthermore, the fact that aberrant splice junctions originate the majority of the tumor-specific peptides identified points to the possibility that deregulation of splicing is playing a significant role in this tumor type as has been described. For example, recurrent mutations in U1 spliceosomal small nuclear RNAs have been associated with SHH medulloblastoma and correlated with changes in splicing[57]. It is therefore plausible to postulate that these unique peptides result from a tumor-specific characteristic, such as aberrant splicing machinery, without the specific events themselves playing a role in tumorigenesis (i.e. passenger events).

In summary, our workflow identifies a robust number of neoantigens sourced from multiple types of tumor-specific genomic and transcriptomic events using very low tissue input and employing native immuno-proteasome processing and presentation machinery to select and expand an autologous personalized T cell therapy. Such a specific, targeted T cell product could make an ideal backbone for the addition of potentiating immunoadjuvants in patients with high-risk cancers such as relapsed/refractory medulloblastoma.

## Methods

**Cell lines and antibodies**. MB002 and MB004 were gifts from Y.J. Cho (Oregon Health and Science University, Portland, OR, United States). D556[58] was a gift from D. Bigner (Duke School of Medicine). D556 and D283 (D. Bigner, ATCC) cell lines were maintained in Eagle's Minimum Essential Medium supplemented with 10% fetal bovine serum and 100 U/mL penicillin and streptomycin (ThermoFisher Scientific). MB002 and MB004 cells were maintained in culture medium with 1:1 DMEM/F12 (Dulbecco's Modified Eagle Medium: Nutrient Mixture F-12) and Neurobasal™-A Medium supplemented with non-essential amino acids, Sodium Pyruvate, HEPES, GlutaMax, B27, EGF, FGF, Heparin, LIF, 10% fetal bovine serum (ATCC), and 100 U/mL penicillin and streptomycin (All from ThermoFisher Scientific). All cell lines were maintained at 37 °C with 5% $CO_2$ in a 95% humidified atmosphere. All established cell lines were verified with STR analysis (GRCF, Johns Hopkins). Antibodies used in this study are listed in Supplementary Data 10.

**Clinical samples**. Medulloblastoma clinical tumor samples are part of the Children's Brain Tumor Network (formerly CBTTC) study cohort CBTTC_0015a. Additional samples were sourced from the Children's National tumor bank. Informed consent of all patients was obtained under a Children's National Medical Center Institutional Review Board and Children's Brain Tumor Network approved protocol. The subjects (or their parents) whose tumor material was deposited in the tissue banks were consented by the Children's National Medical Center Institutional Review Board and/or the Children's Brain Tumor Network approved protocols. Additional information about clinical samples can be found in Supplementary Data 1. For PBMCs, informed consent for the patient 7316-3778 was obtained under a Children's National Medical Center Institutional Review Board. For the cell line work, the PBMCs (including DCs and T cells) were obtained commercially (Stemcell Technologies, Vancouver, Canada).

**DNA/RNA extraction, library preparation, and sequencing**. RNA-seq and WES data from tumor samples were provided by CBTN and it is available upon request through the public project "Pediatric Brain Tumor Atlas: CBTTC" on the Kids First Data Resource Portal [https://kidsfirstdrc.org/] and Cavatica [https://cbtn.org/]. The data is available under restricted access. Access can be obtained by submitting the CBTN Data Access form found in the Kids First Data Resource Portal [https://kidsfirstdrc.org/] or CBTN web [https://cbtn.org/]. For the RNA sequencing of the D556, D283, MB002, and MB004 cell lines, total RNA was extracted from cells using the RNAeasy Mini Kit (Qiagen) according to the manufacturer's protocol. Strand-specific poly-A selected RNA libraries were sequenced on Illumina HiSeq platform with 2 × 150 bp read length to an average of 200 M reads per sample by GENEWIZ.

**Bioinformatics analysis of WGS data**. Somatic variant calling was done following the GATK-Mutect2 best practices[23,59,60]. Briefly, Raw read sequences were mapped to the GRCh38 reference human genome with bwa[61], and duplicates were marked

with PicardTools (v2.18.1). Indel realignment and base recalibration was done with GATKv3.8. Recalibrated reads were used for variant calls using Mutectv2 following default settings and a panel of normal (PON) including the MB germline WGS cohort (Supplementary Data 1). For tumors with no available germline WGS, the Mutectv2 tumor-only mode was used. Only variants on coding sequences were called.

**Bioinformatic analysis of RNA-seq data**. RNA-seq raw reads were mapped to the human reference genome GRCh38 using STAR (v.2.5.1)[62] with 2-pass alignment mode to get better alignment around aberrant splice junctions and Ensembl release 84 annotations (Homo_sapiens.GRCh38.84.gtf). In the case of D556, D283, MB002, and MB004 cell lines, RNA-seq reads were used for variant calls following the GATK best practices for RNA-seq. Briefly, RNA-seq were aligned to the genome using the 2-pass alignment mode of STAR (v.2.5.1). Next, duplicated reads were marked with MarkDuplicates tool (PicardTools). Then, we used the SplitN-CigarReads and IndelRealigner tools from GATKv3.8[23,59,60] to split RNA-seq reads and realign reads around indels. Finally, variants were called using HaplotypeCaller tool (GATKv3.8) and filtered using the VariantFiltration tool (GATK v3.8) with the -window 35 and -cluster 3 options, and Fisher Strand values (FS > 30.0) and Qual by Depth values (QD < 2.0).

**Gene fusion calling**. Gene fusions were called using the following software: Defuse[22], ericscript[21], and STAR-Fusion[20]. Default settings were used in all cases.

**Database generation**. Tumor-specific databases using single nucleotide variations and small insertions/deletions (SNVs), fusions, and tumor-specific isoform variants were generated for each tumor and cell line. *Creation of Variant Peptide Database*: We used the R package CustomProDB[63] to generate variant peptides resulting from DNA SNVs. Coding Mutect2 calls that passed all the filters were incorporated into the genomic sequences and translated to proteins using the Ensembl release 84 transcript annotations. *Creation of Aberrant Splice Junction Peptide database*: To generate protein databases from aberrant splice junctions, we used the R package CustomProDB[63]. This package uses as input a bed12 file with each junction found in the RNA-seq alignment bam file. The junction BED files were derived from RNA-seq alignments using the "junctions extract" function of regtools[64]. This bed12 file contains the chromosome number, the start, and end position of the junction, and the block size of each exon; block sizes are calculated based in the longest read spliced, the standard format for a bed12 file. Then, CustomProDB removes any junction that is annotated in the reference annotation transcript file (ENSEMBL release 84 transcript annotation gtf file) and classifies the junctions into six types of aberrant splice junction: (1) junctions that connect two known exons, with two subtypes: (a) alternative splicing junction where the exons belong to the same gene or (b) fusion, where the exons belong to different genes, (2) junctions that connect a known exon and a region that overlaps with a known exon, (3) junctions that connect a known exon and a non-exon region, (4) junctions that connect two regions overlapping known exons, (5) junction that connect a region overlapping a known exon and a non-exon region, and (6) junctions that connect two non-exon regions. The non-exon regions could be anywhere, e.g. in intronic regions of the same gene, intronic regions of different genes, or intergenic regions. Finally, each putative junction is translated in 3frames using the block size information in the bed file. As an expression cut-off, we required at least 5 reads spanning splice junctions; junctions with less than 5 reads were not included in the database. *Creation of Fusion Peptide database*. Fusion breakpoint coordinates were extracted from the fusion callers and the resulting fused DNA sequences were translated in 6-frames. Stop to stop protein-coding regions with more than 6 consecutive amino acids were included in the database. Finally, tumor-specific SNVs, junctions, and fusion peptides were merged together with human UniProt proteome (UP000005640). The databases generated contain an average of 259,048 entries, 100,179 of them corresponding to the human UniProt proteome database. The average ORF of the databases was 175.8 amino acids. The average ORF of each of the databases (including human UniProt proteome), the average ORF of each of the event types were 21.9, 47.7, and 959 amino acids for fusions, junctions, and SNV, respectively. Detailed information for each tumor database can be found in Supplementary Data 2.

**LC-MS analysis of peptides**. Protein lysates from cellular tumors such as medulloblastoma yield about 10% protein by tissue weight. Importantly when dealing small pediatric tumor samples, the protein input for this pipeline is only 100 mcg, or 1 mg of tissue. As it is also necessary to isolate nucleic acids for RNA/DNA sequencing, 10−15 mg of tissue, roughly the size acquired from a single pass of a stereotactic biopsy needle, is sufficient. The tumor cells were lysed in RIPA buffer (Pierce) by homogenization followed by sonication. The lysate was centrifuged at $20,000 \times g$ for 30 min at 4 °C, and the cleared supernatant collected. The protein concentration was determined by BCA assay (Pierce) and 100 μg of total protein lysate was processed for each sample. The proteins were extracted with methanol:chloroform, air-dried and dissolved in 8M urea followed by dilution to 2M concentration, and digested with sequencing grade LysC enzyme (Thermo Scientific) for 4 h at 37 °C or Trypsin overnight at 37 °C. The resulting peptides were desalted and fractionated into eight fractions using the high-pH fractionation kit (Pierce). The peptide mixtures from each fraction were

sequentially analyzed by LC-MS/MS using the nano-LC system (Easy nLC1000) connected to a Q Exactive HF mass spectrometer (Thermo Scientific). The platform is configured with a nano-electrospray ion source (Easy-Spray, Thermo Scientific), Acclaim PepMap 100 C18 nanoViper trap column (3 µm particle size, 75 µm ID × 20 mm length), and EASY-Spray C18 analytical column (2 µm particle size, 75 µm ID × 500 mm length). The peptides were eluted at a flow rate of 300 nL/min using linear gradients of 7−25% Acetonitrile (in aqueous phase and 0.1% Formic Acid) for 80 min, followed by 45% Acetonitrile for 25 min, and static flow at 90% Acetonitrile for 15 min. The newly generated mass spectrometry proteomics data have been deposited to the ProteomeXchange Consortium via the PRIDE[65] partner repository with the dataset identifier PXD029082

**Mass spectrometry data analysis**. The LC-MS/MS data were collected in data-dependent mode switching between one full scan MS mode (m/z 380-1400, resolution 60 K, AGC 3e6, max ion time 20 ms), and 10 MS/MS scans (resolution 15 K, AGC 1e5, max ion time 120 ms, nCE 27) of the top 10 target ions. The ions were sequenced once and then dynamically excluded from the list for 30 s. The MS raw data sets were analyzed using Thermo Proteome Discoverer Software (version 2.3). The spectrum files were recalibrated using Trypsin or LysC digested indexed Human UniProt database, and peptide spectrums were searched against a tumor-specific custom database using the Sequest HT algorithm at precursor mass tolerance of 10 ppm, and fragment mass tolerance of 0.02 Da. Methionine oxidation and N-terminus acetylation were specified as dynamic modifications. For each tumor, a fully- and a partially-digested search was performed. Peptides and proteins were filtered using a Percolator at a target FDR of 0.01 and a Xcorr > 1.

**Peptide filtering**. A highly stringent filtering strategy was developed in order to filter out previously annotated and unannotated but normal peptides (Fig. S1). This strategy was divided into two steps. (i) To identify previously annotated peptides, we used the BLASTP tool[25] to remove tumor-specific peptides that matched any of the following protein databases: UniProtKB/Swiss-Prot including isoforms, NCBI human non-redundant sequences (including all non-redundant GenBank CDS translations, PDB, SwissProt, PIR, and PRF excluding environmental samples from WGS projects) and neXtProt. (ii) To identify unannotated normal peptides, we used 2 different approaches, one based on proteomics and another based on genomics. First, we performed proteomic profiling of five healthy childhood cerebellum samples using the same methods as the tumor samples. These proteomic raw files were searched against the tumor-specific databases and each non-annotated peptide identified both in the normal cerebellum and in the tumor tissue was removed, leaving only peptides identified exclusively in the tumor tissue. Second, for each peptide identified from fusion or junction events, we evaluated if the fusion or junction events were also detected in a collection of related tissue RNA-seq files from the Genotype-Tissue Expression (GTEx) Project (Supplementary Data 1). For fusion events, the exact breakpoint genomic coordinates in each arm were compared. For example, if we detected a peptide arising from a fusion event with the breakpoint coordinates chr1:15,908,861 and chr5:38,702,49, and if the same breakpoints were detected in any of the GTEx normal tissues analyzed, this peptide was removed. Similarly, for junctions, exact genomic coordinates were compared. For example, a peptide arising from a junction with coordinates chr9:132618441-132642004 would be removed if the same junction is detected in any of the GTEx tissues used. The GTEx data used for the analyses described in this manuscript were obtained from dbGaP accession number phs000424.v2.p1.

**Peptide hydrophobicity index prediction**. Peptide sequence specific hydrophobicity index (HI) was calculated with the SSRCalc vQ tool[27]. The parameters were set to 100 Å C18 column, 0.1% Formic Acid separation system and only unmodified peptides were included. Observed retention times were collected from Proteome Discoverer PSM files. If a peptide was detected multiple times (multiple psm) the average retention time was used. Retention times were plotted against the predicted HI and fitted to a linear model using the R function "lm". R squared and p-value was calculated using the same "lm" function.

**Retention time prediction**. Retention times were predicted using the deep learning algorithm AutoRT[28]. A training model (MB model) was created with all non-modified peptides detected in the cohort using AutoRT default settings. For peptides identified multiple times, an average retention time (RT) was calculated. RT was predicted for 4000 random normal UniProt matched peptides and for all tumor-specific peptides identified in the 46 MB tissues. Alternatively, the model used as an example in the AutoRT publication (model PXD006109, using peptides data from the PXD006109 dataset[29]) was used to calculate RT for a subset of normal peptides matched to UniProt or for all tumor-specific peptides in the 46 MB tumors. Experimental retention times were plotted against the predicted retention times and fitted to a linear model using the R function "lm". R squared and p-value was calculated using the same "lm" function.

**Synthetic peptides**. Peptides for spectra validation and T cell stimulation were synthesized by GenScript with >98% purity and TFA removal.

A common approach to the manufacture of antigen-specific T cells is to identify open reading frames (ORFs), either non-annotated or annotated, and then generate iterative overlapping peptides attempting to find antigenic peptide sequences that will bind to the MHC complex in the context of a particular patient's HLA type. This is advantageous where the same peptide mixture can be used off-the-shelf to manufacture a T cell product for every patient irrespective of their HLA type[34]. However, in order to target a personalized unique antigen set with complete HLA specificity, our approach is to instead identify longer peptides and rely upon autologous antigen-presenting cells to process them into the proper length and sequence to bind MHC I and II molecules in the proper HLA context. Such protocols using DCs pulsed with overlapping peptide pools (15mers overlapping by 11 amino acids) have been used for the manufacture of tumor-associated antigen (TAA) specific T cells that have been used clinically to treat patients with solid tumors. Long synthetic peptides are rapidly and much more efficiently processed by DCs, resulting in an increased presentation to CD4+ and CD8+ T cells. Long synthetic peptides are detected very rapidly in an endolysosome-independent manner after internalization by DCs, followed by proteasome processing, transport, and Ag processing-dependent MHC class I and Class II presentation[51–54].

**Generation of antigen-presenting cells**. Dendritic cells (DC) and peripheral blood mononuclear cells (PBMC) from partially HLA-matched healthy donors were derived from commercially available cryopreserved PBMCs (Stemcell Technologies, Vancouver, Canada) and patients 7316-3778. PBMCs were purified on day 0 by Ficoll density gradient centrifugation (SepMate, Stemcell Technologies, Vancouver, Canada) according to the manufacturer's protocol. Red blood cells were lysed in ACK buffer and PBMCs incubated at 37 °C, 5% CO2 (6 well plate, 4 mL DC medium, 10−15e6/well). After 2 h, the non-adherent fraction was removed by gentle flushing followed by two rinses with PBS to remove lymphocytes. The adherent fraction was cultured in DC medium (CellGenix® GMP DC Medium + 1% L-Glutamine (200 mM) + IL-4 (1000 U/mL; R&D) and GM-CSF (800 U/mL; R&D). The non-adherent cells (NAC) were cryopreserved in freezing medium (50% RPMI-1640, 40% FBS, 10% DMSO). Differentiating monocytes/DCs were fed on day 3 or day 4 by half medium removal and replacement with fresh DC medium + IL-4 and GM-CSF; cytokines at 2X concentration to account for the final volume. On day 7, immature DCs were harvested and incubated with cell line/tumor-specific peptides that had been resuspended at 10 µg/ml in DC medium containing IL-4 and GM-CSF. DCs were resuspended in 100 µL of peptide solution in a 15 mL tube with a loose lid (37 °C, 5% CO2). After 4−6 h, 2.5 mL DC maturation medium was added (CellGenix® GMP DC Medium + L-Glut) + IL-4 + GM-CSF + IL-1β (10 ng/ml) + IL-6 (10 ng/ml) + TNF-α (10 ng/ml) + LPS (30 ng/mL) and DCs were transferred to a 24 well plate for 16−18 h (37 °C, 5% CO2).

**Generation of PHA blasts**. For phytohemagglutinin (PHA) blast generation, PBMC or NAC were stimulated with 5 mg/mL of the mitogen PHA (Sigma-Aldrich) to promote blast formation (PHA blasts). PHA blasts were initially cultured in RPMI-1640 supplemented with 5% human serum (Valley Biomedical, Inc), 2 mmol/L Gluta-Max. Every 1−3 days, PHA blasts were split 1 in 2 and/or fed with a medium containing IL-2 (100 U/mL; R&D).

**Induction of de novo T cell response in vitro**. At a maximum of 16−18 h following maturation, peptide-loaded DCs were irradiated (25 Gy) and co-cultured with non-adherent cells at a ratio of 1:10 in 24-well plates (1e5 DC: 1e6 NAC). T cell medium for stimulation 1 comprised RPMI-1640 (60%) + Click's Medium (40%) + Human serum (10%) + GlutaMax (1%) + IL-6 (100 ng/mL) + IL-7 (10 ng/mL) + IL-12 (10 ng/mL) + IL-15 (5 ng/mL). Cells were fed with complete RPMI-1640 (no cytokines) on day 3 and day 6 if required based on medium color and cell density. On day 6, a fresh batch of DCs was peptide-loaded, matured overnight, and used to restimulate the proliferating cells on day 7. Stimulation 2 medium contained IL-7 (10 ng/mL) and IL-2 (100 U/mL). Restimulation with DCs was repeated on day 14 with a medium containing IL-2 only. On day 21, expanded cells were harvested and used for ELISpot assays and flow cytometric analysis. Cells used for cytotoxicity assays were either plated fresh or following overnight rest after cryopreservation. Cells used for TCR sequencing were cryopreserved.

**Anti-IFN-γ enzyme-linked immuno-spot (ELISpot) assay**. Peptide recognition by expanded cells was assessed by anti-IFN-γ ELISpot. Multi-Screen HTS filter plate membranes (Millipore) were activated with 70% EtOH, washed with PBS, coated with IFN-γ capture antibody (10 mg/mL; Mabtech), and incubated overnight at 4 °C. The next day plates were washed with PBS and blocked with T cell medium for 1 h at 37 °C to control for non-specific protein binding. Expanded cells were washed, resuspended at 1e6 cells/mL in T cell medium, and 5e4−1e5 cells added to appropriate wells, in the presence or absence of peptide-loaded DCs as appropriate. Pooled or single peptides (100 µL; 0.2 mg/mL) were added to appropriate wells along with actin (100 µL; 0.2 mg/mL; irrelevant peptide control) and Staphylococcal enterotoxin B (SEB; positive control). Plates were incubated for 24 h at 37 °C. Plates were developed by washing 6 times in PBS/0.05%Tween 20 followed by incubation with biotinylated IFN-γ detection antibody (0.5 mg/mL; Mabtech; 2 h, 37 °C). This was followed by a further 6 washes in PBS/0.05%Tween

20 and incubation with avidin DH-coupled biotinylated peroxidase H complex (Vectastain Elite ABC Kit; Vector Laboratories) for 1 h in the dark at room temperature (RT). Following three washes in PBST and three washes in PBS, spot formation was detected by incubation with 3-Amino-9-Ethylcarbazole (AEC) substrate for 4 min in the dark. Spot-forming units (SFU) were counted and evaluated by Zellnet Consulting using an automated plate reader system (Zeiss). Recognition of pooled neoantigen peptides and individual peptides was compared against no peptide and actin. In the absence of antigen-presenting cells (APC) a positive T cell response was defined as a minimum of 5 SFU and a fold increase of ≥2.5 over actin. In the presence of APCs, due to increased background, a more stringent positive response was defined as a fold increase of ≥2 over actin or T cell +APC, whichever was higher.

**HLA-blocking experiments.** HLA-restriction of antigen recognition was tested in anti-IFN-γ ELISpot using autologous DCs loaded with the relevant peptide or without peptide (negative control) and blocking antibodies against HLA class I and II (both Dako). DCs were incubated with peptides overnight and the next day anti-HLA mAbs were added to ELISpot plates for 1 h prior to the addition of expanded cells at a ratio of 1 DC to 50 expanded cells. ELISpot plates were incubated and developed as described above.

**T cell phenotyping by flow cytometry.** Freshly harvested expanded cells were washed with PBS, FcR blocked, and stained for viability (LIVE/DEAD™ Fixable Aqua Dead Cell Stain Kit, ThermoFisher Scientific, MA, USA). Cell surface molecules were labeled with optimally titrated mAbs, then cells were fixed (Cytofix, BD Biosciences), permeabilized (Perm Wash, BD Biosciences) and labeled with mAbs against intracellular targets according to the manufacturer's instructions. Cell populations were defined as follows: T central memory (TCM): CD3+ CD4/8+CD45RO+CCR7+CD62L+/−; T effector memory (TEM): CD3+CD4/8+CD45RO+CCR7−CD62L−; T effector (TEFF): CD3+CD4/8+CD45RO−CCR7−CD62L-; putative natural killer cell (NK): CD3− CD56+CD16+/−; putative natural killer T cell (NKT): CD3+CD56+/−; T stem cell memory (TSCM): CD3+CD4/8+CD45RO−CR7+CD62L+CD95+CD127+. The acquisition was performed on a Beckman Coulter CytoFlex S using CytExpert version 2.2.0.97 software. Analysis was performed using FlowJo v10.7.2.

The gating tree for detecting CD3+CD4+CD8+TCRγδ+, $T_{REG}$s, NK and NKT cells in a single panel was as follows (Fig. S10A): 1. Time/FSH (events collected during stable Flow and excludes debris). 2. FSC/SSC (cell distribution light scatter based on size and intracellular composition, respectively) 3. FSC-height/FSC-area (pulse geometry allows exclusion of events that fall outside single-cell range) 4. Live/Dead Aqua/CD3+ (identifies T cells, putative NK cells, and live cells). 5. CD4-BV605/CD8-BV421 (identified CD4+CD8+CD4−CD8−). 6. Gamma delta T cells were classified as CD3+CD4−CD8−TCRγδ+CD56+/− 7. CD4+ cells were further interrogated by bivariate examination of CD25 and CD127 to identify putative $T_{REG}$s, classified as CD3+CD4+CD25+CD127dim. 8. Putative NKT cells were classified simply as CD3+CD56+ and CD8+ T cells were classified as CD3+CD8+CD56−. 9. Putative NK cells were classified as CD3−CD56+CD16+/−. The gating tree for detecting T central memory (TCM), T effector memory (TEM), T effector (TEFF) & T stem cell memory (TSCM) cells in a single panel was as follows (Fig. S10B): 1−5 identical to S10A. Antigen-experienced cells (CD4+/8+CD45RO+) were further assessed as TCM (CD62L +CCR+/− or TEM (CD62L-CCR7-. (Antigen-inexperienced cells (CD45RO−) were further interrogated for TEFF (CD62L−CCR7−) and TSCM (CD62L +CCR7+CD95+CD127+) status. Where > 10% of events fell on an axis bi-exponential scaling was used to visualize all cells on the plot.

**Analysis of IFN-γ and TNF-α release by flow cytometry.** Freshly harvested expanded cells were incubated with pooled medulloblastoma cell line/tumor-specific peptides (2 ng/μL; 100 uL/well) and anti-CD28/CD49 (5 μL/well); (FastImmune, BD Biosciences, CA, USA). Unstimulated cells (anti-CD28/CD49 only), and actin-stimulated (anti-CD28/CD49 + actin 200 ng/well) cells served as negative and irrelevant antigen controls. After 2 h (37 °C, 5% CO₂), the protein transport inhibitor Brefeldin A (Golgi Plug, BD Biosciences, CA, USA) was added to inhibit cytokine release from the cells. Cells were incubated for a further 4 h, then washed with PBS, FcR blocked and stained for viability (LIVE/DEAD™ Fixable Aqua Dead Cell Stain Kit, ThermoFisher Scientific, MA, USA). Cell surface molecules were labeled with optimally titrated mAbs, fixed (Cytofix, BD Biosciences), permeabilized (Perm Wash, BD Biosciences), and labeled with mAbs against intracellular targets according to the manufacturer's instructions. The acquisition was performed on a Beckman Coulter CytoFlex S using CytExpert version 2.2.0.97 software. Analysis was performed using FlowJo v10.7.2.

**Cytotoxicity assay.** Cryopreserved, expanded cells were quickly thawed in a 37 °C water bath, immediately transferred to 10 ml 37 °C RPMI-1640 medium, and centrifuged (RT, 400 × g, 5 min) to remove DMSO from the cell suspension. The cell pellet was resuspended in T cell medium and counted using a Luna dual fluorescence automated cell counter (Logos Biosystems). Cells were rested overnight in T cell medium + IL-2 (100 U/mL) at 37 °C + 5% CO₂. Target medulloblastoma tumor cells were switched to T cell medium (no IL-2) for 96 h prior to

commencement of co-culture with T cells to allow tumor cells to acclimatise to T cell medium. This reduced spontaneous tumor cell death at the 24 h time that resulted from the change of the medium. PHA blasts were used fresh following the 7-day generation.

On the day of co-culture initiation target tumor cells and were stained with Cell Trace Violet (ThermoFisher Scientific) cell labeling dye as follows. Working in a hood with the light off, a 5 mM stock solution of CellTrace™ was prepared immediately prior to use according to the manufacturer's instructions. Target cells were resuspended at 1e6 cells/mL in PBS (no FBS). CellTrace™ solution was added (1 μL/mL) for a final working solution of 5 μM. Cells were briefly vortexed to ensure even staining and incubated for 20 min at 37 °C, protected from light. Following incubation, culture medium (5 X the staining volume) containing at least 1% human serum was added and cells incubated for 5 min to remove any free dye remaining in the solution. Cells were pelleted by centrifugation, resuspended at 1e6 cells/mL in fresh pre-warmed medium, and incubated for at least 10 min to allow cells to undergo acetate hydrolysis before proceeding with co-culture.

While tumor targets were incubating, expanded T cells were pelleted, counted, and resuspended at 1e6 cells/mL in medium+IL-2. T cells were dispensed in triplicate into 4 × 96-well U-bottom plates (D0, 48, and 96 h) in ratio target:TSA-T 1:10.

Following staining, 1e4 target cells were added to the appropriate wells and medium added for a final volume of 200 μL. Targets only were plated as a background viability control. T cell, tumor cell, and blast-only wells were plated for unstained and single stain controls for the viability Cell Trace dyes. OneComp eBeads (Invitrogen) were used for single stain controls for anti-CD3 BV785 (BioLegend; Clone: OKT4).

As soon as possible after plating (D0), and at 48 and 96 h, co-cultures were stained and fixed as follows. For D556 cells, to halt activation and detach adherent cells, 20 μL/well of 20 mM EDTA in PBS was added, mixed by pipetting, incubated for 15 min at room temperature, then mixed again by pipetting to resuspend adhered cells fully. Plates were centrifuged (RT, 400 × g, 4 min), supernatants aspirated and 5 μL FcR block (Miltenyi; Fc Receptor Blocking Reagent human; 130-059-901) added to cells in residual liquid for 5 min to control against the non-specific binding. Without washing 100 μL Fixable Live Dead Green viability dye (1/1000; ThermoFisher Scientific) was added and the cells were incubated for 15 min at RT in the dark. Cells were washed once by adding 100 μL FACS Buffer (PBS + 0.5% FBS) and centrifuging (RT, 400 × g, 4 min). After aspiration, cells were gently pipetted to resuspend the pellet and 50 μL of pre-prepared antibody cocktails added to the residual liquid (1 min, RT, in the dark). FACS Buffer (50 μL) was added to unstained cells. Following AB labeling, cells were washed once by the addition of 100 μL FACS Buffer and centrifugation (RT, 400 × g, 4 min). Cells were fixed by adding 100 μL 4% paraformaldehyde to the residual liquid, incubating at 4 °C for 15 min, and washed by the addition of 100 μL FACS Buffer followed by centrifugation (RT, 400 × g, 4 min). Cells were resuspended in 100 μL FACS Buffer, the plates sealed, foil-wrapped, and stored at 4 °C until acquisition.

**Flow cytometry acquisition and data analysis.** Flow cytometry acquisition was performed on a CytoFlex S (Beckman Coulter) equipped with 405, 488, 561m, and 638 nm lasers and calibrated weekly. The instrument was set to acquire a minimum of 50,000 single, live CD3+ cells on a low flow rate. The time parameter was included to detect fluidics issues during sample acquisition and doublets were excluded using fluorescence peak integral versus height. Spectral overlap was compensated for using beads labeled with the same mAB used in the panel (OneComp Beads, eBioscience), CellTrace-labeled cells (in cytotoxicity assays), and dead cells (viability dye). Post-acquisition compensation was applied and samples analyzed following the application of flow stability, pulse geometry, and viability gates to exclude debris, doublets, and dead cells prior to analysis of cells of interest. Unstained cells, single-stain positive controls, and fluorescence minus one (FMO) controls were used to determine background auto-fluorescence and set positive and negative gates. Data were analyzed using FlowJo version 10.7.2 analysis software (BD Biosciences, Ashland, OR). To determine specific T cell cytokine responses, all samples were compared to the respective controls, and percent positive and/or fold change (FC) was calculated. In samples with low numbers of positive events, a positive response was defined as a fold change ≥ 10. The gating strategy and sample dot plots are shown in Supplementary Fig. 15 (ICS; function), 16a (cell populations/phenotype), 16b (differentiation status), and 18 (cytotoxicity).

**Immunosequencing.** RNA was extracted from peptide-stimulated T cells, using an RNA Easy mini kit, (Qiagen). TCR Vβ CDR3 sequencing was performed by Adaptive Biotechnologies using the survey level resolution Immunoseq platforms (Adaptive Biotechnologies, Seattle, WA). Analysis and compilation of sequence results were performed using the immunoSEQ ANALYZER v3.0 from Adaptive Technologies. TCR-seq data are freely accessible through ImmuneAccess [https://clients.adaptivebiotech.com/pub/rivero-hinojosa-2021-nc]

**Reporting summary.** Further information on research design is available in the Nature Research Reporting Summary linked to this article.

## Data availability

The genomic CBTN data (RNA-seq and WGS) in this study are publicly available. The genomic CBTN data used in this study are available through the public project "Pediatric

Brain Tumor Atlas: CBTTC" on the Kids First Data Resource Portal [https://kidsfirstdrc.org/] and Cavatica [https://cbtn.org/]. The data is available under restricted access. Access can be obtained by submitting the CBTN Data Access form found in the Kids First Data Resource Portal [https://kidsfirstdrc.org/] or CBTN web [https://cbtn.org/]. Additionally, cell line RNA-seq reads are available in the Sequence Read Archive (SRA) under the accession number SRP276163. The newly generated mass spectrometry proteomics data have been deposited to the ProteomeXchange Consortium via the PRIDE[65] partner repository with the dataset identifier PXD029082. TCR-seq data are freely accessible through ImmuneAccess [https://clients.adaptivebiotech.com/pub/rivero-hinojosa-2021-nc]. This work used previously generated genomic data from GTEX project obtained from dbGaP under accession number phs000424.v2.p1. This work used the UniProt human proteome database UP000005640. This work used the previously generated proteomic dataset deposited on PRIDE under the accession number PXD006109. The remaining data are available within the Article, Supplementary Information, or Source Data file. Source data are provided with this paper.

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

## Acknowledgements

The Genotype-Tissue Expression (GTEx) Project was supported by the Common Fund of the Office of the Director of the National Institutes of Health, and by NCI, NHGRI, NHLBI, NIDA, NIMH, and NINDS. This research was conducted using data (WGS and RNA-seq) and tumor specimens made available by The Children's Brain Tumor Network (CBTN). Funding for these experiments was provided by the Chance for Life foundation, The Children's Cancer Foundation, The Jeff Gordon Children's Foundation, The Children's Brain Tumor Foundation, and the Lilabean Foundation.

## Author contributions

Conceived and designed the experiments: S.R.-H., M.G., C.M.B. and B.R.R. Performed the experiments: M.G., A.P., H.Z. and V.C. Analyzed the data: S.R.-H., M.G. and A.P. Wrote the paper: S.R.-H., M.G., C.M.B. and B.R.R. All authors read and approved the final manuscript.

## Competing interests

The authors declare no competing interests.
