## [Peer Review File · Nature Communications]

Reviewers' Comments:

Reviewer #1:

Remarks to the Author:

The manuscript by Rivero-Hinojosa et al. describes a proteogenomic approach that identifies novel tumour-associated protein products in medulloblastoma (MB). Authors show that they can find similar protein products with their approach in established MB tumour cell lines. They are able to expand a T cell clone with one of the proteomics peptides identified, and show that this clone can recognise the peptide when pulsed on DC and efficiently and selectively kills the cell line.

The study is certainly relevant, but there are important considerations that I have summarised in my comments below.

Main Comments

1. The workflow described here does not identify HLA-associated peptides that can be recognised by T cells, but rather detects proteins that are expressed and detectable in tumours encoded by non-conventional sources. This needs to be absolutely clear throughout, and the motivation stated by the authors that a combined Lys-C and trypsin digest could somehow represent HLA-presented peptides is – apologies for being blunt – upsetting. HLA-bound peptides have many different properties beyond just their length, and antigen processing and presentation is a complex and yet not fully characterised cellular process that is subject to a whole field of research. Specifically the peptide C-termini, which will be exclusively basic (K and R) in this study, prohibits the statement that they are in any way comparable to HLA-presented sequences. Instead, authors should state what they have actually done, which is performed a careful, but standard proteogenomics study.
2. A justification would be needed for pulsing the digested peptide sequences on DCs over designing overlapping peptide libraries covering novel confirmed open reading frames(ORFs), which would be the more systematic approach.
3. Much more detail is needed in the main results text to understand how exactly novel protein sequences were mapped. Which genes are these sequences derived from? What chromosomal rearrangements were observed? The methods part is missing the number of entries and average length of ORFs included in the proteomic searches.
4. I do not think it is justified to claim tumour-specificity over tumour-association for the identified ORFs expressed. A lot more information is needed in my view to claim a tumour-specific event. Again, more details are needed in the result text. From the methods I take it that 5 healthy cerebellum samples were analysed, and those sequences that were present in these tissues were removed. What were the exact numbers here? Was the majority of peptides removed and deemed non-specific? What was the variability between samples and how do you know that you have enough coverage to claim tumour-specificity? Have you considered the limit of detection in your mass spectrometry experiments? Did you remove ORFs detected in GTEx healthy tissues as well for the tumour data, or only for cell line data?

Minor Comments

Title: "Low-input" can be misinterpreted without context – please remove.

Line 22: Is "incorporate" the correct word here?

Line 45/46: "...TAA....approaches may be limited by central tolerance..." Should clarify that this is only a limitation if vaccination and/or stimulation is used, however, no limitation with recombinant T cell receptor or MHC-peptide specific antibody approaches.

Line 159 "synthetic versions of the 7316-3778 tumor peptides " Could you please clarify that number is the patient ID (I was first confused), and state how many peptides were validated? "genomic events/proteomics events" axis labelling does not make sense. Is it a 'proteomic' event, or a protein product of a genomic event?

Reviewer #2:

Remarks to the Author:

Rivero-Hinojosa and co-authors describe their concept, workflow and demonstration of a

proteogenomic analysis capable of defining tumor specific peptide neoantigens which can prime an anti-tumor T-cell response. The majority of peptides targeted are as a result of splice junction and as such medulloblastoma is a good model for which to test such a strategy given its relatively low mutational burden. Whilst the concept of using novel or aberrant splice junctions as the basis of an immune-therapy is not in itself novel the authors have made a very good first step at tackling how this may be achieved in practice and have dealt with some of the inherent technical challenges. I regard this as a good proof-of concept with some good potential for future application and I think relevant not just for medulloblastoma but for anyone operating in the immunogenomics sphere in tumors with low mutation rates (i.e. pediatrics).

There are a few points that I feel should probably be addressed further where possible.

In the abstract the authors refer to the resultant personalised T cell therapies as highly-specific and it is the specificity which I think could do with some more fleshing out.

For example, "peptides that originate from novel genomic events also present in normal healthy tissues from the Genotype-Tissue Expression (GTEx) project" were removed. It's not self evident to me from this statement or from the description in the methods what this involves? Could this be explained better (and the whole informatics pipeline really)? Also it's clear that even after applying the GTEx filter for example on subject (7316-3778) at least some of the peptides can also be identified in the healthy cerebellum proteome? How do the authors account for this? To what extent is this likely to cause a problem? Presumably the idea for the future is delivery in patients. Overlap with the healthy cerebellum is one thing but what about overlap with other normal tissues?

When the authors say the "vast majority" of unique novel peptides discovered are non-overlapping between the cell lines how many overlapping are they talking about and could they therefore clarify what the term unique novel peptide means in this context? The same question applies to the analysis of the 42 primary tissues. If these are aberrant splices are they not induced by tumorigenic processes and could we not expect overlap and commonality?

What is the nature of the novel splice junctions? There isn't a description or comment of how these are typically formed? Is there any pattern to them for instance are they all runons? What type of transcripts in what arrangement tend to produce these peptides and Is there some reason they would only show up in cancers?

Is there any sense about any cellular heterogeneity or plasticity in the expression of these novel splice junctions which might affect future therapeutic application? When 15% of cells remain after treatment with TSAT is there any difference in the expression of these novel junctions in the remaining cells? Differential cell survival?

Maybe I'm missing something as to why this can't be done but if the claims of highly personalised and specific therapy are true and unique novel peptides are majority non-overlapping wouldn't a good control be to prime with the MB002 unique peptides and apply TSAT to D556 and vice versa i.e. testing cross-reactivity if donors can be matched to one line can be matched to another possibly.

Reviewer #3:

Remarks to the Author:

Rivero-Hinojosa et al., presents a proteogenomics study to identify neoantigen peptides in pediatric brain tumors. They used RNA-Seq, whole genome and LC-MS/MS data from 46 tumor tissues to identify tumor-specific SNVs, novel junctions and fusion genes that are also found at the protein level and show an example in which a tumor-specific peptide is able to incorporate MHC II-based T cell responses and are cytolytic against tumors in vitro. Overall, I found this a very well done, interesting and important study demonstrating the promise of personalized treatment based on proteogenomic investigation.

Major:

- As an additional filtering step I would also use the neXtProt database
- I like the way that the authors validated the retention time and spectral match after filtering out normal human peptides. However, it was unclear to me whether or not the peptides that did not correlate well were filtered out of the downstream analysis, especially those that have markedly different predicted and experimental values.
- Was there a threshold for peptide spectral matches to determine if a peptide was a true positive?
- It's quite rare to identify fusions at the protein level. Did all of these peptides span the fusion junction? Or are they frameshifts of one of the fusion genes? It would be useful to show this in some way in the supplement so the reader can better understand exactly where these peptides are deriving from.
- Are the novel splice junction peptides alternative splicing of known exons, in intronic regions, in non-coding regions? I am assuming this information was used to determine the number of frameshifts for the database search and therefore would include information about this in STable 3 and 7.
- What is happening with 7316-3544, 7316-777 and 7316-906? They seem to have much higher junctions/fusions at the protein level that's not reflected at the genome level.
- Were there any novel peptides that were identified in both cells lines and human tissue? Or novel peptides consistently found in many of the patient samples?

Minor:

- I suggest changing instances of 'mass spec' to MS or LC-MS/MS
- Your samples all start with '7316' – I would suggest removing this suffix and keep the final 4 digits as identifiers to make the plots easier to read.
- Figure 3A formatted looked off in my version
- Missing a ')' on line 177
- The green and blue colors used in Figures 4 are hard to distinguish

We thank the Reviewers for their valuable comments and the modifications they suggested as well as the opportunity to submit a revised version. Every point raised by the referees has been addressed by a response and an addition or amendment to the text where relevant to the question raised (please see our point-by-point response to the Reviewers below). We believe the paper has been markedly strengthened and hope that it is now acceptable for publication.

REVIEWER COMMENTS.

Reviewer #1 (Remarks to the Author): with expertise in immunopeptidomics.

The manuscript by Rivero-Hinojosa et al. describes a proteogenomic approach that identifies novel tumour-associated protein products in medulloblastoma (MB). Authors show that they can find similar protein products with their approach in established MB tumour cell lines. They are able to expand a T cell clone with one of the proteomics peptides identified, and show that this clone can recognise the peptide when pulsed on DC and efficiently and selectively kills the cell line.

The study is certainly relevant, but there are important considerations that I have summarised in my comments below.

Main Comments:

1. The workflow described here does not identify HLA-associated peptides that can be recognised by T cells, but rather detects proteins that are expressed and detectable in tumours encoded by non-conventional sources. This needs to be absolutely clear throughout, and the motivation stated by the authors that a combined Lys-C and trypsin digest could somehow represent HLA-presented peptides is – apologies for being blunt - upsetting. HLA-bound peptides have many different properties beyond just their length, and antigen processing and presentation is a complex and yet not fully characterised cellular process that is subject to a whole field of research. Specifically the peptide C-termini, which will be exclusively basic (K and R) in this study, prohibits the statement that they are in any way comparable to HLA-presented sequences. Instead, authors should state what they have actually done, which is performed a careful, but standard proteogenomics study.

RESPONSE: *We agree with the reviewer on this point and wish to clarify that our primary objective is not the direct identification of HLA-associated peptides. We apologize for the confusion and have clarified the overall goal of the work throughout the manuscript. We also completely agree that trypsin/LysC digested peptides do not recapitulate HLA-presented peptides and recognize that some sentences in the text can be interpreted in that way. For instance, the justification for the usage of LysC instead of trypsin for protein digestion is not to increase identification of HLA-presented peptides or that the identified peptides are more comparable to HLA-presented sequences. Instead, the justification is that LysC yields longer peptides that, after processing by dendritic cells into the relevant HLA binding sequence, can potentially bind either MHC Class I or II. Essentially, pulsing the DCs with longer peptides gives them more options in selecting HLA-compatible epitopes. This has been clarified in the text (lines 103-111) and throughout the manuscript.*

2. A justification would be needed for pulsing the digested peptide sequences on DCs over designing overlapping peptide libraries covering novel confirmed open reading frames (ORFs), which would be the more systematic approach.

RESPONSE: *We thank the Reviewer for their comment. In regard to pulsing DCs, several publications have shown that DCs can efficiently intake, process, and present long synthetic peptides¹⁻⁴. Such protocols using DCs pulsed with overlapping peptide pools (15mers overlapping by 11 amino acids) have also been used for the manufacture of tumor associated antigen (TAA) specific T cells that have been used clinically to treat patients with solid tumors⁵. Long synthetic peptides are rapidly and much more efficiently processed by DCs, resulting in an increased presentation to CD4⁺ and CD8⁺ T cells. Long synthetic peptides are detected very rapidly in an endolysosome-independent manner after internalization by DCs, followed by proteasome processing, transport, and Ag processing-dependent MHC class I and Class II presentation. This approach allows us to reduce cost of peptides libraries compared to multiple overlapping shorter (8-9 aa) pepmixes from a few ORFs.*

Our GMP-compliant manufacturing protocol using overlapping peptide pools (pepmixes) is our standard approach for the generation of antigen-specific T cells where the same pepmix (or pepmixes) can be used “off the shelf” to manufacture a T cell product for every patient irrespective of their HLA type. Specifically, our TAA-T cell therapy protocols that are currently being evaluated clinically utilize TAA pepmixes targeting WT1, survivin, and PRAME and contain over 220 peptides for these 3 protein targets⁵. (However, when applied to a personalized, patient-specific clinical setting where the vast majority of the proteogenomically identified neoantigens are unique to an individual tumor, clinical grade peptides need to be synthesized for each individual patient. Thus, these expensive reagents (~\$1200 per peptide) must necessarily be limited in number and we are most confident in those peptides which have been identified by MS rather than those that have only been predicted by a novel ORF. In this way, we are also able to target the peptide products from multiple novel ORFs providing greater insurance against heterogeneity. This justification has been included in the methods section (lines 593-606) and discussion lines (418-421).

3. Much more detail is needed in the main results text to understand how exactly novel protein sequences were mapped. Which genes are these sequences derived from? What chromosomal rearrangements were observed? The methods part is missing the number of entries and average length of ORFs included in the proteomic searches.

RESPONSE: *We apologize for the lack of clarity. Extended information (genes, chromosomal coordinates, ORF, sequence, etc.) has been added to Supplementary dataset 3 (tumor tissues) and Supplementary dataset 7 (cell lines). Additional information about the databases, number of entries and ORF lengths has been added to the Methods section (lines 518-522) and Supplementary dataset 2. Briefly, our databases contain an average of 259,048 entries, 100,179 of them correspond to the human UniProt proteome database. The average ORF of the databases is 175.8 amino acids (including human UniProt proteome), the average ORF of each of the event types were 21.9, 47.7 and 959 amino acids for fusions, novel junctions and SNV, respectively. Detailed information for each tumor can be found in Supplementary dataset 2.*

4. I do not think it is justified to claim tumour-specificity over tumour-association for the identified ORFs expressed. A lot more information is needed in my view to claim a tumour-specific event. Again, more details are needed in the result text. From the methods I take it that 5 healthy cerebellum samples were analysed, and those sequences that were present in these tissues were removed. What were the exact numbers here? Was the majority of peptides removed and deemed non-specific? What was the variability between samples and how do you know that you have enough coverage to claim tumour-specificity? Have you considered the limit of detection in your mass spectrometry experiments? Did you remove ORFs detected in GTEx healthy tissues as well for the tumour data, or only for cell line data?

RESPONSE: *We thank the Reviewer for their important comments and questions. We have clarified that we consider a peptide to be tumor-specific when the peptide is identified exclusively in the tumor. Our pipeline is designed to filter out potential normal/non-annotated peptides as well as to eliminate possible alignment/calling errors made by the NGS read aligners and/or fusion/splicing callers. We used both proteomic and genomic (RNA-seq) techniques for the filtration steps. First, we generated proteomic data (peptide spectra) from 5 healthy childhood cerebellar tissues, the tissue of origin for medulloblastoma, using the same sample prep and mass spec protocols as we used for the tumor tissues and cell lines. Then, the healthy cerebellum MS/MS spectra were searched against each individual personalized database from the medulloblastoma tumors. Every non-annotated peptide identified in both the normal cerebella and the tumor tissue was removed, leaving only novel peptides identified exclusively in the tumor tissue. As a second filtration step, for each novel peptide identified from a fusion or junction event, we evaluated if the fusion or junction event was also detected in a collection of related tissue RNA-seq from GTEX (Supplementary dataset 1). For fusion events, the exact breakpoint genomic coordinates in each arm were compared. For example, if we detected a peptide arising from a fusion event with the breakpoints coordinates chr1:15,908,861 and chr5:38,702,49, and found the same breakpoints in any of the GTEX normal tissues analyzed, that peptide was removed. Similarly, for junctions, exact junctional genomic coordinates were compared. For example, a peptide arising from a junction with coordinates chr9:132618441-132642004 would be removed if the same junction was detected in any of the GTEX tissues used. These filtration steps were done for both the medulloblastoma tissues and cell lines.*

Regarding the number of peptides eliminated in each step we have added a new figure depicting the number of peptides removed in tumor tissues in each step (Figure 2c). Overall, 481 novel peptides were identified, 17 (3.53 %) peptides were removed as they were present also in normal cerebellar tissues, 102 (21%) peptides were removed as the same genomic event that originated that peptide were detected in the GTEX RNA-seq collection, leaving 362 unique novel peptides. As we now show in Supplementary dataset 3, expressed as per tumor averages, 13.56 peptides were identified (range 2-56), 1.5 peptides (range 0-6) were removed as they were present also in normal cerebellar tissues, 3.6 peptides (range 0-13) were removed because the same genomic event that originated that peptide were detected in the GTEX RNA-seq collection. A mean of 9 novel tumor specific peptides were identified per tumor. Figure 2c illustrates the number of peptides removed from each tumor tissue. The exact same approach was use in cell lines

and is illustrated in Figure 4c. The relatively small number of peptides removed by the filtration steps is consistent with a stringent process that limits false discoveries. In addition to the Supplementary dataset 3, this information has been added in the text (lines 130-134). Detailed explanation about the filtration steps has been added to the Materials and Methods section (lines 561-571).

We also agree with the reviewer that detection limit and coverage are important factors in defining a tumor specific event. To control for coverage, we have used 5 healthy cerebellar tissues, the tissue which medulloblastoma arises. These samples were prepared and run on LC-MS/MS with the same protocol resulting in a similar number of scans in the normal cerebellum files compared to tumor data, as shown in the figure below. While we cannot absolutely discard the possibility that some of the tumor-specific peptides can be found in other normal cells, such an occurrence appears to be very unlikely due to the low number of peptides removed using the normal cerebellum proteomic filter (i.e. only 3.53% of the peptides were removed in that step).

Number for scans in proteomic raw files from healthy cerebellum and medulloblastoma tumor tissue. The coverage in both types of samples is comparable.

Additionally, we wanted to clarify that we are using “tumor specific antigen” as a term-of-art meant to convey a distinction from “tumor associated antigens” which are antigens known to be expressed on normal tissues as well. It is not meant to carry a connotation of absolute tumor specificity, rather a relative restriction to tumor tissue. For example, most TSA are the result of missense mutations which are usually tumor restricted⁶. However, it is understood that normal tissues can contain somatic mutations that are not tumorigenic in all tissue contexts. The same has also been shown for specific oncogenic fusions such as *ews-fli1* which also require cooperating enhancer genotypes⁷.

Minor Comments

- Title: "Low-input" can be misinterpreted without context – please remove.

RESPONSE: *We have removed “Low-input” from the title.*

- Line 22: Is "incorporate" the correct word here?

RESPONSE: *The word “incorporate” has been replaced by “generate”.*

- Line 45/46: "...TAA....approaches may be limited by central tolerance...." Should clarify that this is only a limitation if vaccination and/or stimulation is used, however, no limitation with recombinant T cell receptor or MHC-peptide specific antibody approaches.

RESPONSE: *We thank the Reviewer for highlighting this important point. We have modified that sentence to clarify that it only pertains to ex vivo expanded T cells (lines 39-40).*

- Line 159 "synthetic versions of the 7316-3778 tumor peptides " Could you please clarify that number is the patient ID (I was first confused), and state how many peptides were validated?

"genomic events/proteomics events" axis labelling does not make sense. Is it a 'proteomic' event, or a protein product of a genomic event?

RESPONSE: *We apologize for the confusion. 7316-3778 is the patient ID. We have modified that sentence for clarity and added that 7 tumor peptides from the patient ID 7316-3778 were validated, now shown in Supplementary Figure 6 and included in the text (line 162).*

“Proteomic event” indicates events detected at protein/peptide level (identified by mass spectrometry) while “Genomic Event” indicates events detected at the genomic level (identified by RNA-seq/WGS). As we are using Genomic events to create our protein databases, a Proteomic event is always matched to a Genomic event as well. However Genomic events do not have to be detected at the proteomic level, which is the case for most of them. We have changed the Y-axis labels and used this terminology in the figure legend of Figures 2 and 4 for better understanding.

Reviewer #2 (Remarks to the Author): with expertise in medulloblastoma and brain tumors – immunogenomics.

Rivero-Hinojosa and co-authors describe their concept, workflow and demonstration of a proteogenomic analysis capable of defining tumor specific peptide neoantigens which can prime an anti-tumor T-cell response. The majority of peptides targeted are as a result of splice junction and as such medulloblastoma is a good model for which to test such a strategy given its relatively low mutational burden. Whilst the concept of using novel or aberrant splice junctions as the basis of an immune-therapy is not in itself novel the authors have made a very good first step at tackling how this may be achieved in practice and have dealt with some of the inherent technical challenges. I regard this as a good proof-of concept with some good potential for future application and I think relevant not

just for medulloblastoma but for anyone operating in the immunogenomics sphere in tumors with low mutation rates (i.e. pediatrics).

There are a few points that I feel should probably be addressed further where possible.

1) In the abstract the authors refer to the resultant personalised T cell therapies as highly-specific and it is the specificity which I think could do with some more fleshing out.

For example, "peptides that originate from novel genomic events also present in normal healthy tissues from the Genotype-Tissue Expression (GTEx) project" were removed. It's not self evident to me from this statement or from the description in the methods what this involves? Could this be explained better (and the whole informatics pipeline really)? Also it's clear that even after applying the GTEx filter for example on subject (7316-3778) at least some of the peptides can also be identified in the healthy cerebellum proteome? How do the authors account for this? To what extent is this likely to cause a problem? Presumably the idea for the future is delivery in patients. Overlap with the healthy cerebellum is one thing but what about overlap with other normal tissues?

RESPONSE: *We thank the Reviewer for their comments and questions and we apologize for the lack of clarity. Our filtration strategy only applies to novel junction and fusion events as the somatic mutation pipeline (GATK best practices) already includes extensive filtering steps to remove known SNPs, by filtering out any SNPs annotated in The Genome Aggregation Database (gnomAD), and germline mutations from germline WGS samples used in this study.*

Critically, the filtration approach in our pipeline is designed to filter out potential novel normal/non-annotated peptides as well as to eliminate possible alignment/calling errors made by the NGS read aligners and/or fusion/splicing callers. We used both proteomic and genomic (RNA-seq) techniques for the filtration steps. First, we generated proteomic data (peptide spectra) from 5 healthy childhood cerebellar tissues, the tissue of origin for medulloblastoma, using the same sample prep and mass spec protocols as we used for the tumor tissues and cell lines. Then, the healthy cerebellum MS/MS spectra were searched against each individual personalized database from the medulloblastoma tumors. Every non-annotated peptide identified in both the normal cerebella and the tumor tissue was removed, leaving only novel peptides identified exclusively in the tumor tissue. As a second filtration step, for each novel peptide identified from a fusion or junction event, we evaluated if the fusion or junction event was also detected in a collection of related tissue RNA-seq from GTEx (Supplementary dataset 1). For fusion events, the exact breakpoint genomic coordinates in each arm were compared. For example, if we detected a peptide arising from a fusion event with the breakpoints coordinates chr1:15,908,861 and chr5:38,702,49, and found the same breakpoints in any of the GTEx normal tissues analyzed, that peptide was removed. Similarly, for junctions, exact junctional genomic coordinates were compared. For example, a peptide arising from a junction with coordinates chr9:132618441-132642004 would be removed if the same junction was detected in any of the GTEx tissues used. These filtration steps were done both for medulloblastoma tissues and for cell lines. Additional description of the filtration steps has now been included in the methods (lines 561-571) and results sections (lines 130-135).

2) When the authors say the "vast majority" of unique novel peptides discovered are non-overlapping between the cell lines how many overlapping are they talking about and could they therefore clarify what the term unique novel peptide means in this context? The same question applies to the analysis of the 42 primary tissues. If these are aberrant splices are they not induced by tumorigenic processes and could we not expect overlap and commonality?

RESPONSE: We thank the Reviewer for their question and apologize for the confusion. The "unique novel peptides" refer to novel peptides that have been identified only in one patient or a single cell line. For cell lines, as shown in the revised Figure 4C, only 5 out of 269 peptides (1.86%) were identified in more than one cell line. Moreover, no peptides were found to be in common between cell lines and primary tumor tissues. In primary tumors, as shown in revised Figure 4C, 18 out of 362 peptides (4.97%) were identified in more than one tumor; 12 of these (3.3%) were found in 2 tumors and 6 (1.6%) were found in 5 or more tumors (see Figure below and new Supplementary Figure 7).

Frequent novel peptides found in 46 medulloblastoma tumors.

Only one peptide was identified in more than 20% of the samples (12 out of 46). All of these shared peptides resulted from novel splice junctions. For example, peptide NSSVSGIFTFQK can arise from different novel junction events in the DDX31 gene. We also detected novel alternative splices between exon 14 and exons 17, 18 and 19. In

addition, we detected a novel junction between intron 13 and the exon 14. This novel junction changes the frame of exon 14, originating peptide NSSVSGIFTFQK. Interestingly, the splicing between exon 14 and exon 17, 18 and 19 returns to the annotated frame for DDX31. The annotated splicing between exons 14 and 15 will introduce a stop codon as a consequence of this change in the frame of exon 14. We have included this information in the text (lines 166-186) and in the new Supplementary Figure 7.

Further, DDX31 is a DEAD-box RNA helicase conserved across eukaryotes, but it has not been well-studied. DDX31 was found to be mutated in several group 4 medulloblastomas in a previous sequencing study⁸. Complex rearrangement and focal deletions of the DDX31 gene have also been observed in several Group 4 medulloblastomas; these deletions occur concurrently with amplification of the OTX2 locus, a known medulloblastoma oncogene^{9,10}. This finding suggests that DDX31 mutation (either by deletion or truncation) may cooperate with the oncogenic role of OTX2. Linking our findings to previously published work, such a deletion or rearrangement could originate the NSSVSGIFTFQK peptide by modifying the splicing partner of DDX31. The role of these isoforms in medulloblastoma is a very interesting topic for future investigation, however we considered it to be outside of the scope of this paper and thus hadn't included it. We have also found novel peptides arising from novel junctions in the CARF, EEA1, LMNB1, LIZIC and VANGL2 genes with a lower frequency, 5 out of 46 tumors. In response to the reviewer's request, we have now included a description of the frequent peptides in the revised Results text (lines 166-186) as well as in a new supplementary figure (Supplementary Figure 7).

As indicated by the reviewer, the uniqueness of the majority of novel peptides indicates that the genomic events that generate these peptides are unlikely to be tumorigenic. Although the number of frequent peptides is low, we cannot disregard the fact that some of these peptides may contribute to cancer as very few frequent driver events have been identified in medulloblastoma tumors despite intensive genomic study. The driver events that have been identified, occur at a relatively low frequency compared with other cancers, particularly adult cancers. The DDX31 finding discussed above may be one such event. It is therefore plausible to postulate that these unique novel peptides result from a tumor specific characteristic, such as aberrant splicing machinery, without the specific events themselves playing a role in tumorigenesis (i.e. passenger events). We have added this point to the discussion (lines 424-436).

3) What is the nature of the novel splice junctions? There isn't a description or comment of how these are typically formed? Is there any pattern to them for instance are they all tumors? What type of transcripts in what arrangement tend to produce these peptides and Is there some reason they would only show up in cancers?

RESPONSE: We thank the Reviewer for their concerns. We have added additional information in the methods section regarding the splice junctions. To generate protein databases from novel splice junction, we used the R package CustomProDB¹¹. This package uses as input a bed12 file with each junction found in the RNA-seq alignment bam file. This bed12 file contains the chromosome, the start and end positions of the junction and the block size of each exon. The block sizes are calculated based on the longest read spliced, as is the standard format for a bed12 file. Then, CustomProDB

removes any junction that is annotated in the reference annotation transcript file (Ensembl release 84 transcripts annotation file) and classifies the junctions as one of 6 types (see figure below):

1. The junction connects two known exons. This can be subdivided in two subtypes: a) novel alternative splicing junction, if the exons belong to the same gene or b) fusion, if the exons belong to different genes (not showed in the figure as these events are infrequent).
2. The junction connects a known exon and a region that overlaps with known exon.
3. The junction connects a known exon and a non-exon region.
4. The junction connects two regions overlapping with known exons.
5. The junction connects a region that overlaps with a known exon with a non-exon region.
6. The junction connects two non-exon regions. The non-exon regions could be anywhere, including intronic regions of the same gene, intronic regions of different genes and intergenic regions.

Finally, each putative novel junction is translated in 3-frames using the block size information in the bed file. As an expression cut off, we required at least 5 reads spanning a splice junction, i.e. any novel junction with less than 5 reads was not included in the database. We have included additional information about the junction types in the methods section (lines 494-516).

Figure. Junction types used.

As indicated in the figure above, the largest number of peptides detected from novel junctions originated from junctions that connect two non-exon regions (40%). As many long non coding RNAs are not annotated in the reference annotation files, and several^{12,13} reports have shown that non-coding RNA can be a source of neoantigens in tumors, we used the latest non-coding database from the NONCODE project (www.noncode.org) to whether any of our novel peptides could be generated from non-coding RNA. We found that 21% of the peptides generated by non-exon region junctions overlapped with non-coding RNAs in the human Noncodev6 database, suggesting that a portion of that category of peptides originated from non-coding RNAs. Some proteomic studies have reported the detection of peptides from non-coding RNAs^{12,13} and their potential use as neoantigens, however, to our knowledge, the mechanism behind their translation has not been described. We attempted to link the remaining 79% to other genomic features such as transposable elements, repeated elements, etc., but did not find any linkage. In addition, we have not discovered any common characteristics in the genes that originated those peptides.

The deregulation of splicing in cancer has been describe before. For example, recurrent mutations in U1 spliceosomal small nuclear RNAs has been associated with SHH

medulloblastoma and correlated with changes in splicing¹⁴. Although, it would certainly be very interesting to study the mechanisms that originate those novel splicing events, doing so would be outside the scope of this report which seeks instead to look toward clinical translation rather than biological mechanism.

We wish to thank the reviewer for this suggestion and have updated the discussion to include these important points raised by the Reviewer (lines 407-412).

4) Is there any sense about any cellular heterogeneity or plasticity in the expression of these novel splice junctions which might affect future therapeutic application? When 15% of cells remain after treatment with TSAT is there any difference in the expression of these novel junctions in the remaining cells? Differential cell survival?

RESPONSE: *The reviewer has proposed a very interesting line of investigation that bears directly on the clinical translation of this work. It is very likely that tumor cell sub-populations manifest different genomic events giving rise to different neoantigens. Indeed, the problem of antigen heterogeneity confounds many T cell therapy approaches including CAR-T and is the primary reason why our approach was designed to target multiple neoantigens. It is hoped that epitope spreading will also help to guard against the escape of individual clones that under-express the targeted antigens by engaging endogenous T cells specific for additional liberated antigens in the inflammatory milieu. In exploring this idea further, several limitations unfortunately arise. We have attempted to create individualized proteogenomic databases using single cell RNA-seq, however the low coverage of current single cell RNA-seq techniques is insufficient to identify a significant number of novel splicing events. Another limitation is that the establishment of short-term cultures from a patient's tumor tissue is complicated by high rates of initial cell death which would confound the type of "treatment resistance" experiments one could envision to answer these questions. In the cell line application, cells remaining after TSA-T cytotoxicity experiments may have survived due to mis-presentation of their unique clonal antigens due to HLA mismatch. Thus, the results would likely misrepresent the outcome.*

5) Maybe I'm missing something as to why this can't be done but if the claims of highly personalized and specific therapy are true and unique novel peptides are majority non-overlapping wouldn't a good control be to prime with the MB002 unique peptides and apply TSAT to D556 and vice versa i.e. testing cross-reactivity if donors can be matched to one line can be matched to another possibly.

RESPONSE: *We agree with the Reviewer that this is a logical line of reasoning and is indeed one that we pursued in the ELISPOT antigen specificity experiments. As shown in Figure 5, panels a and b, T cells stimulated with MB002 peptides are not activated in response to dendritic cells loaded with D556 peptides or Actin control peptides. Similarly, MB002 T cells do not respond to D556 peptides as shown in the intracellular flow cytometric staining for IFN- γ and TNF- α in Figure 5d. The reverse experiments showing that D556 specific T cells do not respond to MB002 peptides are showed in the Supplementary Figures 12, 13 and 14.*

Unfortunately, HLA differences between the cell lines and the PBMC donors prevent us from extending this analysis to the cytotoxicity experiments due to both allogeneic killing and the lack of epitope recognition resulting from HLA mismatch, either of which would

confound the results. In the context of clinical translation, the T cell products will be autologous so HLA mismatch will not be an issue.

Reviewer #3 (Remarks to the Author): with expertise in proteogenomics

Rivero-Hinojosa et al., presents a proteogenomics study to identify neoantigen peptides in pediatric brain tumors. They used RNA-Seq, whole genome and LC-MS/MS data from 46 tumor tissues to identify tumor-specific SNVs, novel junctions and fusion genes that are also found at the protein level and show an example in which a tumor-specific peptide is able to incorporate MHC II-based T cell responses and are cytolytic against tumors in vitro. Overall, I found this a very well done, interesting and important study demonstrating the promise of personalized treatment based on proteogenomic investigation.

Major:

1) As an additional filtering step I would also use the neXtProt database Same as uniprot?

RESPONSE: *We thank the Reviewer for their comment. As suggested, we filtered our data using the neXProt database (line 129 and 557). This step did not remove any additional peptide in any sample, as all entries present in neXProt are also present in either the Uniprot Proteome database or other databases used in the filtering steps.*

2) I like the way that the authors validated the retention time and spectral match after filtering out normal human peptides. However, it was unclear to me whether or not the peptides that did not correlate well were filtered out of the downstream analysis, especially those that have markedly different predicted and experimental values.

RESPONSE: *We apologize for the confusion. We did not set a specific cutoff value for the exclusion of individual peptides. The envisioned purpose of the retention time validation was to evaluate the proteogenomic pipeline by testing the null hypothesis that the discovered novel peptides are simply inaccurate spectral matches. Incidentally, we did check and none of the novel peptides from the 3716-3778 sample had an RT that varied by more than 2 SD from the modeled RT.*

3) Was there a threshold for peptide spectral matches to determine if a peptide was a true positive?

RESPONSE: *We used the Proteome Discoverer Xcorr value to provide a reasonable determination of whether the spectral match was a true positive. The Xcorr is a search-dependent score. It scores the number of fragment ions that are common to two different peptides with the same precursor mass and calculates the cross-correlation score for all candidate peptides queried from the database by SEQUEST searches. Peptides with a Xcorr<1 were removed. Additionally, only peptides with a 1% FDR were considered significant. This information has been included in the Methods section (lines 549-540).*

4) It's quite rare to identify fusions at the protein level. Did all of these peptides span the fusion junction? Or are they frameshifts of one of the fusion genes? It would be useful to show this in some way in the supplement so the reader can better understand exactly where these peptides are deriving from.

RESPONSE: We thank the Reviewer for their comments and agree that the detection of peptides spanning a fusion is rare in proteomics. Further, recurrent fusions are rare in medulloblastoma and none of them have been found to be biologically functional. We did not detect any peptides that span a fusion between two canonical protein sequences; all of them involve a frameshift of one gene. Also, exon-exon fusions are not the most common, with most of the fusions taking place in the 3'UTR, intronic and intergenic regions. We have added this information to Supplementary dataset 3 and Supplementary dataset 7 as suggested by the reviewer.

5) Are the novel splice junction peptides alternative splicing of known exons, in intronic regions, in non-coding regions? I am assuming this information was used to determine the number of frame-shifts for the database search and therefore would include information about this in Table S3 and 7.

RESPONSE: We thank the Reviewer for raising this question and agree this information could be valuable to the reader and have included it in the Supplementary dataset 3 and Supplementary dataset 7.

6) What is happening with 7316-3544, 7316-777 and 7316-906? They seem to have much higher junctions/fusions at the protein level that's not reflected at the genome level.

RESPONSE: The reviewer is correct that these 3 samples exhibit a unique disconnect between proteomic and genomic events. We examined correlations between the molecular subgroup, shared mutations, and types of junctions present in these samples but find no correlations. Of course, this analysis is limited by the small sample size but at this time, we are unable to explain the finding.

7) Were there any novel peptides that were identified in both cells lines and human tissue? Or novel peptides consistently found in many of the patient samples?

RESPONSE: We did not identify any common peptides between the cell lines and the tumors. For cell lines, as shown in the revised Figure 4c, only 5 out of 269 peptides (1.86%) were found in more than one cell line. It is of note that no peptides were found to be in common between cell lines and tumor tissues. In tumors, as observed in the revised Figure 4C, 18 out of 362 peptides (4.97%) were found in more than one tumor; 12 of these (3.3%) were found in just 2 tumors and 6 (1.6%) were found in 5 or more tumors (Figure below and new Supplementary Figure 7).

Only one peptide was found in more than 20% of the samples (12 out of 46). All these shared peptides result from novel splice junctions. The peptide NSSVSGIFTFQK can arise from different novel junction events in the DDX31 gene. We detect novel alternative splices between exon 14 and exons 17, 18 and 19. In addition, we detect a novel junction between intron 13 and the exon 14. This novel junction changes the frame of exon 14, originating peptide NSSVSGIFTFQK. Interestingly, the splicing between exon 14 and exon 17, 18 and 19 returns to the annotated frame for DDX31. The annotated splicing between exons 14 and 15 will introduce a stop codon as a consequence of this change in the frame of exon 14.

Frequent novel peptides found in 46 medulloblastoma tumors.

DDX31 is a DEAD-box RNA helicase conserved across eukaryotes, but it has not been well-studied. DDX31 was found to be mutated in several group 4 medulloblastomas in a previous sequencing study⁸. Complex rearrangement and focal deletions of the DDX31 gene have also been observed in several Group 4 medulloblastomas; these deletions occur concurrently with amplification of the OTX2 locus, a known medulloblastoma oncogene^{9,10}. This finding suggests that DDX31 mutation (either by deletion or truncation) may cooperate with the oncogenic role of OTX2. Linking our findings to previously published work, such a deletion or rearrangement could originate the NSSVSGITTFQK peptide by modifying the splicing partner of DDX31. The role of these isoforms in medulloblastoma is a very interesting topic for future investigation, however we considered it to be outside of the scope of this paper and thus hadn't included it. We have also found novel peptides arising from novel junctions in the CARF, EEA1, LMNB1, LZIC and VANGL2 genes with a lower frequency, 5 out of 46 tumors. In response to the reviewer's request, we have now included a description of the frequent peptides in the revised text (lines 166-186) as well as a new supplementary figure (Supplementary Fig. 7).

Minor:

- I suggest changing instances of 'mass spec' to MS or LC-MS/MS.

RESPONSE: *We agree with the Reviewer and have replaced mass spec with LC-MS/MS in the text and figures.*

- Your samples all start with '7316' – I would suggest removing this suffix and keep the final 4 digits as identifiers to make the plots easier to read.

RESPONSE: *The figures have been updated for better visualization. Following reviewer's recommendation, we have removed the 7316 suffixes in all figures.*

- Figure 3A formatted looked off in my version.

RESPONSE: *The figures have been updated for better visualization.*

- Missing a ')' on line 177

RESPONSE: *We have added the parenthesis to the line 177*

- The green and blue colors used in Figures 4 are hard to distinguish

RESPONSE: *We thank the Reviewer for their comments. The colors have been changed through the figures, especially trying to eliminate displaying green and blue together.*

BIBLIOGRAPHY

- 1 Bijker, M. S. *et al.* Superior induction of anti-tumor CTL immunity by extended peptide vaccines involves prolonged, DC-focused antigen presentation. *Eur J Immunol* **38**, 1033-1042, doi:10.1002/eji.200737995 (2008).
- 2 Ma, W. *et al.* Long-Peptide Cross-Presentation by Human Dendritic Cells Occurs in Vacuoles by Peptide Exchange on Nascent MHC Class I Molecules. *J Immunol* **196**, 1711-1720, doi:10.4049/jimmunol.1501574 (2016).
- 3 Menager, J. *et al.* Cross-presentation of synthetic long peptides by human dendritic cells: a process dependent on ERAD component p97/VCP but Not sec61 and/or Derlin-1. *PLoS one* **9**, e89897, doi:10.1371/journal.pone.0089897 (2014).
- 4 Rosalia, R. A. *et al.* Dendritic cells process synthetic long peptides better than whole protein, improving antigen presentation and T-cell activation. *Eur J Immunol* **43**, 2554-2565, doi:10.1002/eji.201343324 (2013).
- 5 Hont, A. B. *et al.* Immunotherapy of Relapsed and Refractory Solid Tumors With Ex Vivo Expanded Multi-Tumor Associated Antigen Specific Cytotoxic T Lymphocytes: A Phase I Study. *Journal of clinical oncology : official journal of the American Society of Clinical Oncology* **37**, 2349-2359, doi:10.1200/JCO.19.00177 (2019).
- 6 Wojas-Krawczyk, K. & Kubiakowski, T. Imperfect Predictors for Lung Cancer Immunotherapy-A Field for Further Research. *Frontiers in oncology* **10**, 568174, doi:10.3389/fonc.2020.568174 (2020).
- 7 Johnson, K. M., Taslim, C., Saund, R. S. & Lessnick, S. L. Identification of two types of GGAA-microsatellites and their roles in EWS/FLI binding and gene regulation in Ewing sarcoma. *PLoS one* **12**, e0186275, doi:10.1371/journal.pone.0186275 (2017).
- 8 Robinson, G. *et al.* Novel mutations target distinct subgroups of medulloblastoma. *Nature* **488**, 43-48, doi:10.1038/nature11213 (2012).
- 9 Adamson, D. C. *et al.* OTX2 is critical for the maintenance and progression of Shh-independent medulloblastomas. *Cancer research* **70**, 181-191, doi:10.1158/0008-5472.CAN-09-2331 (2010).
- 10 Di, C. *et al.* Identification of OTX2 as a medulloblastoma oncogene whose product can be targeted by all-trans retinoic acid. *Cancer research* **65**, 919-924 (2005).
- 11 Wang, X. & Zhang, B. customProDB: an R package to generate customized protein databases from RNA-Seq data for proteomics search. *Bioinformatics* **29**, 3235-3237, doi:10.1093/bioinformatics/btt543 (2013).
- 12 Chong, C. *et al.* Integrated proteogenomic deep sequencing and analytics accurately identify non-canonical peptides in tumor immunopeptidomes. *Nat Commun* **11**, 1293, doi:10.1038/s41467-020-14968-9 (2020).
- 13 Laumont, C. M. *et al.* Noncoding regions are the main source of targetable tumor-specific antigens. *Science translational medicine* **10**, doi:10.1126/scitranslmed.aau5516 (2018).
- 14 Suzuki, H. *et al.* Recurrent noncoding U1 snRNA mutations drive cryptic splicing in SHH medulloblastoma. *Nature* **574**, 707-711, doi:10.1038/s41586-019-1650-0 (2019).

Reviewers' Comments:

Reviewer #1:

Remarks to the Author:

The authors have comprehensively answered my queries, with a few minor outstanding comments from my side.

I noticed that the wording "low-input" is still being used throughout the manuscript and can, without context, be easily misinterpreted (authors used 10-15mg tissue!). Please state the exact amounts in results and/or methods - it is only mentioned in the introduction and discussion as far as I can see.

The authors strongly justify the proteomics over ligandomics approach, but sufficient discussion on the limitations of the approach are still missing. Limitations are (1) that the MS evidence is limited to presence of protein in the cell and surface presentation needs to be verified, (2) that DC pulsing predominantly results in class II presentation and evaluation of class I presentation is limited, and (3) tumour-specific antigens from unstable antigen sources will be missed. A very short discussion on these limitations would be useful.

I feel uncomfortable with the sentence "...dependent upon the use of HLA binding prediction algorithms which have been shown to suffer from a high degree of inaccuracy" and reference used here. Could authors put this into context, please: HLA binding predictors have a high level of accuracy in predicting whether a peptide binds to a specific HLA allele or not, but lack accuracy of predicting which sequence is presented from a protein coding region.

Reviewer #2:

Remarks to the Author:

I won't summarise my opinion or the results and significance to the field again as this is a re-review of a revised manuscript and substantially the results and conclusions are the same, as is my overall positive opinion of the work. The manuscript I think has been strengthened and with respect to my own comments I'm content that the authors have made a good attempt to address them.

With one small exception part of my original 1st comment was "Also it's clear that even after applying the GTEX filter for example on subject (7316-3778) at least some of the peptides can also be identified in the health cerebellum proteome? How do the authors account for this? To what extent is this likely to cause a problem? Presumably the idea for the future is delivery in patients. Overlap with the healthy cerebellum is one thing but what about overlap with other normal tissues?".

Perhaps I am missing this but although the description of the filtering steps has been improved this question has not been addressed directly. I don't know that this is something that should impede publication but on the other hand this isn't something that the authors need to be shy about addressing in the discussion.

In summary, I can see the limitations and technical challenges inherent in the process and those which will come in future usage, but overall I think this is an important first step and a good proof of concept.

Reviewer #3:

Remarks to the Author:

The authors have addressed all of my concerns.

Reviewer #1 (Remarks to the Author):

The authors have comprehensively answered my queries, with a few minor outstanding comments from my side.

I noticed that the wording "low-input" is still being used throughout the manuscript and can, without context, be easily misinterpreted (authors used 10-15mg tissue!). Please state the exact amounts in results and/or methods - it is only mentioned in the introduction and discussion as far as I can see.

Response: We thank the reviewer for pointing out that low-input is a relative term and could be misinterpreted by the reader. The small amount of tissue required for our pipeline is a key aspect of its applicability for brain tumors where small tissue yield from resection is standard unlike many somatic tumors. In the experimental setting, we of course generated an excess of protein lysate to allow for experimentation and therefore the exact amounts used are not informative to this point. Medulloblastoma, a very cellular tumor, yields about 10% protein by tissue weight using our lysis protocol; i.e. 1mg of tissue yields 100 mcg of protein lysate, which is the input amount for our protocol. We also use tissue to generate the nucleic acids for RNA and DNA sequencing. Thus, 10-15 mg of tissue, roughly the size of a piece of cooked long grain rice, is sufficient. This can be acquired from a single pass of a stereotactic biopsy needle and is thus applicable to even the unresectable brain tumor given adequate cellularity. This is contrast to published ligandomic reports, for example in melanoma (another cellular tumor), in which starting tissue ranged from 100 to 4000 mg (Bassani-Sternberg, M. et al. Direct identification of clinically relevant neoepitopes presented on native human melanoma tissue by mass spectrometry. Nat Commun 7, 13404 (2016)). We feel that the above explanation is perhaps excessive for the published work and have added an abbreviated explanation in the methods section. If the editor would prefer, we can of course expand upon the point.

Added to methods under clinical tumor samples: Protein lysates from cellular tumors such as medulloblastoma yield about 10% protein by tissue weight. The input for the proteomic portion of this pipeline is 100 mcg, or 1 mg of tissue. As it is also necessary to isolate nucleic acids for RNA/DNA sequencing, 10-15 mg of tissue, roughly the size acquired from a single pass of a stereotactic biopsy needle, is sufficient.

The authors strongly justify the proteomics over ligandomics approach, but sufficient discussion on the limitations of the approach are still missing. Limitations are (1) that the MS evidence is limited to presence of protein in the cell and surface presentation needs to be verified, (2) that DC pulsing predominantly results in class II presentation and evaluation of class I presentation is limited, and (3) tumour-specific antigens from unstable antigen sources will be missed. A very short discussion on these limitations would be useful.

Response: We agree with the reviewer that we can better explain the limitations of this approach. The first limitation noted is correct and we have added it to the manuscript. The second noted limitation is not as clear. While it is true that ELISPOTs demonstrating antigen specificity for TSA discovered from the cell lines showed MHC II specificity, the T cell products created against the cell lines' and the patient's TSAs had equal CD4+ and CD8+ representation (29% CD4+, 31% CD8+ for the patient). In any event, this is a very small sample size from which to draw such a conclusion and it will be investigated in a future planned clinical trial. Further, we are unaware of literature evidence that pulsed DCs in general result in class II presentation. Of course, while they do bear MHC II and thus activate CD4+ T cells, it is in addition to MHC I and thus they are capable of activating CD8+ cells as well. The third point raised regarding missing unstable antigen sources, while potentially limiting, there is nothing precluding the discovery of peptides arising from unstable translation as these should be present in the tumor lysate and subjected to mass spec identification - provided they contain a lysine and are long enough to yield a 6 aa peptide after digestion. This does however highlight the limitation that our workflow requires a lysine to be present for digestion and subsequent search identification and we have added this to the manuscript. In contrast, ligandomics does not use enzyme digestion, replacing this limitation with the high false discovery rate inherent in digestion free searches.

I feel uncomfortable with the sentence "...dependent upon the use of HLA binding prediction algorithms which have been shown to suffer from a high degree of inaccuracy" and reference used here. Could authors put this into context, please: HLA binding predictors have a high level of accuracy in predicting whether a peptide binds to a specific HLA allele or not, but lack accuracy of predicting which sequence is presented from a protein coding region.

Response: The reference cited is a ligandomics paper that plotted their mass spec identified peptides by predicted binding affinity against a constellation of peptides predicted from variant calling. The conclusion drawn was that "The potential and promise of MS detection of neoantigens is in fact highlighted by the hit rate of mutated peptide ligands... This indicates a clear advantage compared with the usage of prediction software tools to identify neoantigens."

We agree with the reviewer's point that criticism of binding prediction is best understood in context, namely that of predicting presentation of antigens originating from predicted novel open reading frames. We presented it in that context but also in a discussion of ligandomics where prediction algorithms are not a required part of the process. We have removed the above sentence from that latter section.

Reviewer #2 (Remarks to the Author):

I won't summarise my opinion or the results and significance to the field again as this is a

re-review of a revised manuscript and substantially the results and conclusions are the same, as is my overall positive opinion of the work. The manuscript I think has been strengthened and with respect to my own comments I'm content that the authors have made a good attempt to address them.

With one small exception part of my original 1st comment was "Also it's clear that even after applying the GTEX filter for example on subject (7316-3778) at least some of the peptides can also be identified in the healthy cerebellum proteome? How do the authors account for this? To what extent is this likely to cause a problem? Presumably the idea for the future is delivery in patients. Overlap with the healthy cerebellum is one thing but what about overlap with other normal tissues?".

Perhaps I am missing this but although the description of the filtering steps has been improved this question has not been addressed directly. I don't know that this is something that should impede publication but on the other hand this isn't something that the authors need to be shy about addressing in the discussion.

In summary, I can see the limitations and technical challenges inherent in the process and those which will come in future usage, but overall I think this is an important first step and a good proof of concept.

Response: *We apologize for overlooking that critique and not adequately responding to it. While our workflow improves upon previous ones regarding efforts to screen out unannotated normal proteins, it cannot be claimed that this process will detect all such proteins. We believe this to be a minor limitation in the context of antigen specific T cell targeting insofar as tumor associated antigens (TAAs) have been targeted with this platform without significant toxicity. TAAs are expressed in limited tissues or at low levels somatically and would thus targeting them would be expected to be analogous to targeting an unannotated normal protein. In addition, central tolerance would be expected to cull any high affinity T cell receptors for such normal proteins. This limitation has been added to the manuscript.*

Reviewer #3 (Remarks to the Author):

The authors have addressed all of my concerns.